# BoTorch: A Framework for Efficient Monte-Carlo Bayesian Optimization

**Maximilian Balandat**
Facebook
balandat@fb.com

**Brian Karrer**
Facebook
briankarrer@fb.com

**Daniel R. Jiang**
Facebook
drjiang@fb.com

**Samuel Daulton**
Facebook
sdaulton@fb.com

**Benjamin Letham**
Facebook
bletham@fb.com

**Andrew Gordon Wilson**
New York University
andrewgw@cims.nyu.edu

**Eytan Bakshy**
Facebook
ebakshy@fb.com

## Abstract

Bayesian optimization provides sample-efficient global optimization for a broad range of applications, including automatic machine learning, engineering, physics, and experimental design. We introduce BoTorch, a modern programming framework for Bayesian optimization that combines Monte-Carlo (MC) acquisition functions, a novel sample average approximation optimization approach, auto-differentiation, and variance reduction techniques. BoTorch's modular design facilitates flexible specification and optimization of probabilistic models written in PyTorch, simplifying implementation of new acquisition functions. Our approach is backed by novel theoretical convergence results and made practical by a distinctive algorithmic foundation that leverages fast predictive distributions, hardware acceleration, and deterministic optimization. We also propose a novel "one-shot" formulation of the Knowledge Gradient, enabled by a combination of our theoretical and software contributions. In experiments, we demonstrate the improved sample efficiency of BoTorch relative to other popular libraries.

## 1 Introduction

Computational modeling and machine learning (ML) have led to an acceleration of scientific innovation in diverse areas, ranging from drug design to robotics to material science. These tasks often involve solving time- and resource-intensive global optimization problems to achieve optimal performance. Bayesian optimization (BO) [75, 46, 76], an established methodology for sample-efficient sequential optimization, has been proposed as an effective solution to such problems, and has been applied successfully to tasks ranging from hyperparameter optimization [24, 92, 110], robotic control [15, 5], chemical design [36, 60, 111], and tuning and policy search for internet-scale software systems [4, 58, 57, 23]. Meanwhile, ML research has been undergoing a revolution driven largely by new programming frameworks and hardware that reduce the time from ideation to execution [43, 16, 1, 81]. While BO has become rich with new methodologies, today there is no coherent framework that leverages these computational advances to simplify and accelerate BO research in the same way that modern frameworks have for deep learning. In this paper, we address this gap by introducing BoTorch, a modular and scalable Monte Carlo (MC) framework for BO that is built around modern paradigms of computation, and theoretically grounded in novel convergence results. Our contributions include:

- A novel approach to optimizing MC acquisition functions that effectively combines with deterministic higher-order optimization algorithms and variance reduction techniques.

- The first convergence results for sample average approximation (SAA) of MC acquisition functions, including novel general convergence results for SAA via randomized quasi-MC.
- A new, SAA-based "one-shot" formulation of the Knowledge Gradient, a look-ahead acquisition function, with improved performance over the state-of-the-art.
- Composable model-agnostic abstractions for MC BO that leverage modern computational technologies, including auto-differentiation and scalable parallel computation on CPUs and GPUs.

We discuss related work in Section 2 and then present the methodology underlying BOTORCH in Sections 3 and 4. Details of the BOTORCH framework, including its modular abstractions and implementation examples, are given in Section 5. Numerical results are provided in Section 6.

## 2 Background and Related Work

In BO, we aim to solve $\max_{x \in \mathbb{X}} f_{\text{true}}(x)$, where $f_{\text{true}}$ is an expensive-to-evaluate function and $\mathbb{X} \subset \mathbb{R}^d$ is a feasible set. BO consists of two main components: a *probabilistic surrogate model* of the observed function—most commonly, a Gaussian process (GP)—and an *acquisition function* that encodes a strategy for navigating the exploration vs. exploitation trade-off [92]. Taking a model-agnostic view, our focus in this paper is on MC acquisition functions.

Popular libraries for BO include Spearmint [94], GPyOpt [98], Cornell-MOE [106], RoBO [52], Emukit [97], and Dragonfly [49]. We provide further discussion of these packages in Appendix A. Two other libraries, ProBO [72] and GPFlowOpt [55], are of particular relevance. ProBO is a recently suggested framework[1] for using general probabilistic programming in BO. While its model-agnostic approach is similar to ours, ProBO, unlike BOTORCH, does not benefit from gradient-based optimization provided by differentiable programming, or algebraic methods designed to exploit GPU acceleration. GPFlowOpt inherits support for auto-differentiation and hardware acceleration from TensorFlow [via GPFlow, 64], but unlike BOTORCH, it does not use algorithms designed to specifically exploit this potential. Neither ProBO nor GPFlowOpt naturally support MC acquisition functions. In contrast to all existing libraries, BOTORCH is a modular programming framework and employs novel algorithmic approaches that achieve a high degree of flexibility and performance.

The MC approach to optimizing acquisition functions has been considered in the BO literature to an extent, typically using stochastic methods for optimization [100, 106, 109, 104]. Our work takes the distinctive view of sample average approximation (SAA), an approach that combines sampling with deterministic optimization and variance reduction techniques. To our knowledge, we provide the first theoretical analysis and systematic implementation of this approach in the BO setting.

## 3 Monte-Carlo Acquisition Functions

We begin by describing a general formulation of BO in the context of MC acquisition functions. Suppose we have collected data $\mathcal{D} = \{(x_i, y_i)\}_{i=1}^n$, where $x_i \in \mathbb{X}$ and $y_i = f_{\text{true}}(x_i) + v_i(x_i)$ with $v_i$ some noise corrupting the true function value $f_{\text{true}}(x_i)$. We allow $f_{\text{true}}$ to be multi-output, in which case $y_i, v_i \in \mathbb{R}^m$. In some applications we may also have access to distributional information of the noise $v_i$, such as its (possibly heteroskedastic) variance. Suppose further that we have a probabilistic surrogate model $f$ that for any $\mathbf{x} := \{x_1, \ldots, x_q\}$ provides a distribution over $f(\mathbf{x}) := (f(x_1), \ldots, f(x_q))$ and $y(\mathbf{x}) := (y(x_1), \ldots, y(x_q))$. We denote by $f_{\mathcal{D}}(\mathbf{x})$ and $y_{\mathcal{D}}(\mathbf{x})$ the respective *posterior* distributions conditioned on data $\mathcal{D}$. In BO, the model $f$ traditionally is a GP, and the $v_i$ are assumed i.i.d. normal, in which case both $f_{\mathcal{D}}(\mathbf{x})$ and $y_{\mathcal{D}}(\mathbf{x})$ are multivariate normal. The MC framework we consider here makes no particular assumptions about the form of these posteriors.

The next step in BO is to optimize an acquisition function evaluated on $f_{\mathcal{D}}(\mathbf{x})$ over the *candidate set* $\mathbf{x}$. Following [105, 7], many acquisition functions can be written as

$$\alpha(\mathbf{x}; \Phi, \mathcal{D}) = \mathbb{E}\big[a(g(f(\mathbf{x})), \Phi) \,|\, \mathcal{D}\big], \tag{1}$$

where $g : \mathbb{R}^{q \times m} \to \mathbb{R}^q$ is a (composite) *objective function*, $\Phi \in \mathbf{\Phi}$ are parameters independent of $\mathbf{x}$ in some set $\mathbf{\Phi}$, and $a : \mathbb{R}^q \times \mathbf{\Phi} \to \mathbb{R}$ is a *utility function* that defines the acquisition function.

In some situations, the expectation over $f_{\mathcal{D}}(\mathbf{x})$ in (1) and its gradient $\nabla_{\mathbf{x}}\alpha(\mathbf{x}; \Phi, \mathcal{D})$ can be computed analytically, e.g. if one considers a single-output ($m=1$) model, a single candidate ($q=1$) point $x$, a

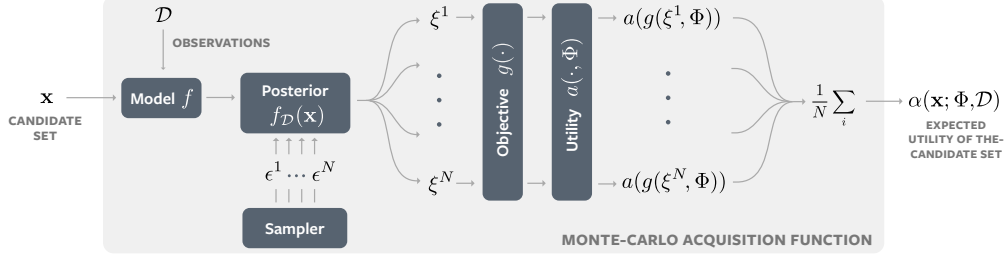

Figure 1: MC acquisition functions. Samples $\xi_{\mathcal{D}}^i$ from the posterior $f_{\mathcal{D}}(\mathbf{x})$ provided by the model $f$ at $\mathbf{x}$ are evaluated in parallel and averaged as in (2). All operations are fully differentiable.

Gaussian posterior $f_{\mathcal{D}}(x) = \mathcal{N}(\mu_x, \sigma_x^2)$, and the identity objective $g(f) \equiv f$. Expected Improvement (EI) is a popular acquisition function that maximizes the expected difference between the currently observed best value $f^*$ (assuming noiseless observations) and $f$ at the next query point, through the utility $a(f, f^*) = \max(f - f^*, 0)$. EI and its gradient have a well-known analytic form [46].

In general, analytic expressions are not available for arbitrary objective functions $g(\cdot)$, utility functions $a(\cdot, \cdot)$, non-Gaussian model posteriors, or collections of points $\mathbf{x}$ which are to be evaluated in a parallel or asynchronous fashion [32, 94, 106, 100, 104]. Instead, MC integration can be used to approximate the expectation (1) using samples from the posterior. An MC approximation $\hat{\alpha}_N(\mathbf{x}; \Phi, \mathcal{D})$ of (1) using $N$ samples $\xi_{\mathcal{D}}^i(\mathbf{x}) \sim f_{\mathcal{D}}(\mathbf{x})$ is straightforward:

$$\hat{\alpha}_N(\mathbf{x}; \Phi, \mathcal{D}) = \frac{1}{N}\sum_{i=1}^{N} a(g(\xi_{\mathcal{D}}^i(\mathbf{x})), \Phi). \tag{2}$$

The obvious way to evaluate (2) is to draw i.i.d. samples $\xi_{\mathcal{D}}^i(\mathbf{x})$. Alternatively, randomized quasi-Monte Carlo (RQMC) techniques [14] can be used to significantly reduce the variance of the estimate and its gradient (see Appendix E for additional details).

## 4 MC Bayesian Optimization via Sample Average Approximation

To generate a new candidate set $\mathbf{x}$, one must optimize the acquisition function $\alpha$. Doing this effectively, especially in higher dimensions, typically requires using gradient information. For differentiable analytic acquisition functions (e.g. EI, UCB), one can either manually implement gradients, or use auto-differentiation to compute $\nabla_x \alpha(x; \Phi, \mathcal{D})$, provided one can differentiate through the posterior parameters (as is the case for Gaussian posteriors).

### 4.1 Optimizing General MC Acquisition Functions

An unbiased estimate of the MC acquisition function gradient $\nabla_{\mathbf{x}} \alpha(\mathbf{x}; \Phi, \mathcal{D})$ can often be obtained from (2) via the reparameterization trick [50, 85]. The basic idea is that $\xi \sim f_{\mathcal{D}}(\mathbf{x})$ can be expressed as a suitable (differentiable) deterministic transformation $\xi = h_{\mathcal{D}}(\mathbf{x}, \epsilon)$ of an auxiliary random variable $\epsilon$ independent of $\mathbf{x}$. For instance, if $f_{\mathcal{D}}(\mathbf{x}) \sim \mathcal{N}(\mu_{\mathbf{x}}, \Sigma_{\mathbf{x}})$, then $h_{\mathcal{D}}(\mathbf{x}, \epsilon) = \mu_{\mathbf{x}} + L_{\mathbf{x}}\epsilon$, with $\epsilon \sim \mathcal{N}(0, I)$ and $L_{\mathbf{x}} L_{\mathbf{x}}^T = \Sigma_{\mathbf{x}}$. If $a(\cdot, \Phi)$ and $g(\cdot)$ are differentiable, then $\nabla_{\mathbf{x}} a(g(\xi), \Phi) = \nabla_g a \nabla_\xi g \nabla_{\mathbf{x}} h_{\mathcal{D}}(\mathbf{x}, \epsilon)$.

Our primary methodological contribution is to take a sample average approximation [53] approach to BO. The conventional way of optimizing MC acquisition functions of the form (2) is to re-draw samples from $\epsilon$ for each evaluation and apply stochastic first-order methods such as Stochastic Gradient Descent (SGD) [105]. In our SAA approach, rather than re-drawing samples from $\epsilon$ for each evaluation of the acquisition function, we draw a set of base samples $E := \{\epsilon^i\}_{i=1}^N$ once, and hold it fixed between evaluations throughout the course of optimization (this can be seen as a specific incarnation of the method of common random numbers). Conditioned on $E$, the resulting MC estimate $\hat{\alpha}_N(\mathbf{x}; \Phi, \mathcal{D})$ is deterministic. We then obtain the candidate set $\hat{\mathbf{x}}_N^*$ as

$$\hat{\mathbf{x}}_N^* \in \arg\max_{\mathbf{x} \in \mathbb{X}^q} \hat{\alpha}_N(\mathbf{x}; \Phi, \mathcal{D}). \tag{3}$$

The gradient $\nabla_{\mathbf{x}} \hat{\alpha}_N(\mathbf{x}; \Phi, \mathcal{D})$ can be computed as the average of the sample-level gradients, exploiting auto-differentiation. We emphasize that whether this average is a "proper" (i.e., unbiased, consistent) estimator of $\nabla_{\mathbf{x}} \alpha(\mathbf{x}; \Phi, \mathcal{D})$ is irrelevant for the convergence results we will derive below.

While the convergence properties of MC integration are well-studied [14], the respective literature on SAA, i.e., convergence of the *optimizer* (3), is far less comprehensive. Here, we derive what, to the best of our knowledge, are the first SAA convergence results for (RQ)MC acquisition functions in the context of BO. To simplify our exposition, we limit ourselves to GP surrogates and i.i.d. base samples; more general results and proofs are presented in Appendix D. For notational simplicity, we will drop the dependence of $\alpha$ and $\hat{\alpha}_N$ on $\Phi$ and $\mathcal{D}$ for the remainder of this section. Let $\alpha^* := \max_{\mathbf{x} \in \mathbb{X}^q} \alpha(\mathbf{x})$, and denote by $\mathcal{X}^*$ the associated set of maximizers. Similarly, let $\hat{\alpha}_N^* := \max_{\mathbf{x} \in \mathbb{X}^q} \hat{\alpha}_N(\mathbf{x})$. With this we have the following key result:

**Theorem 1.** *Suppose (i) $\mathbb{X}$ is compact, (ii) $f$ has a GP prior with continuously differentiable mean and covariance functions, and (iii) $g(\cdot)$ and $a(\cdot, \Phi)$ are Lipschitz continuous. If the base samples $\{\epsilon^i\}_{i=1}^N$ are i.i.d. $\mathcal{N}(0,1)$, then (1) $\hat{\alpha}_N^* \to \alpha^*$ a.s., and (2) $\mathrm{dist}(\hat{\mathbf{x}}_N^*, \mathcal{X}^*) \to 0$ a.s.. Under additional regularity conditions, (3) $\forall \delta > 0$, $\exists K < \infty$, $\beta > 0$ s.t. $\mathbb{P}\big(\mathrm{dist}(\hat{\mathbf{x}}_N^*, \mathcal{X}^*) > \delta\big) \leq K e^{-\beta N}, \forall N \geq 1$.*

Under relatively weak conditions,[2] Theorem 1 ensures not only that the optimizer $\hat{\mathbf{x}}_N^*$ of $\hat{\alpha}_N$ converges to an optimizer of the true $\alpha$ with probability one, but also that the convergence (in probability) happens at an exponential rate. We stated Theorem 1 informally and for i.i.d. base samples for simplicity. In Appendix D.3 we give a formal statement, and extend it to base samples generated by a family of RQMC methods, leveraging recent theoretical advances [79]. While at this point we do not characterize improvements in theoretical convergence rates of RQMC over MC for SAA, we observe empirically that RQMC methods work remarkably well in practice (see Figures 2 and 3).

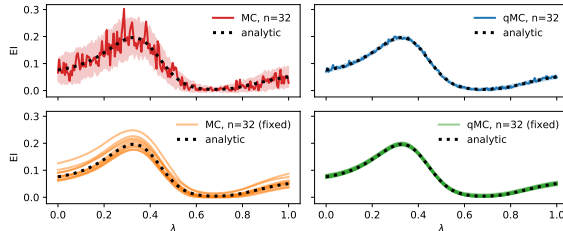
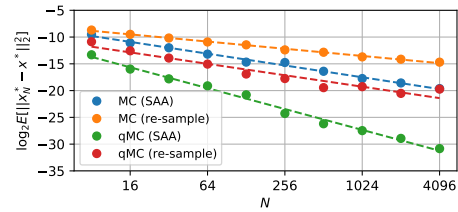

Figure 2: MC and RQMC acquisition functions, with and without ("fixed") re-drawing base samples between evaluations. The model is a GP fit on 15 points randomly sampled from $\mathbb{X} = [0,1]^6$ and evaluated on the Hartmann6 function along the slice $x(\lambda) = \lambda \mathbf{1}$.

Figure 3: Empirical convergence rates of the optimizer for EI using MC / RQMC sampling under SAA / stochastic optimization ("re-sample"). Appendix E provides additional detail and discussion.

The primary benefit from SAA comes from the fact that in order to optimize $\hat{\alpha}_N(\mathbf{x}; \Phi, \mathcal{D})$ for fixed base samples $E$, one can now employ the full toolbox of deterministic optimization, including quasi-Newton methods that provide faster convergence speeds and are generally less sensitive to optimization hyperparameters than stochastic first-order methods. By default, we use multi-start optimization via L-BFGS-B in conjunction with an initialization heuristic that exploits fast batch evaluation of acquisition functions (see Appendix F.1). We find that in practice the bias from using SAA only has a minor effect on the performance relative to using the analytic ground truth, and often improves performance relative to stochastic approaches (see Appendix E), while avoiding tedious tuning of optimization hyperparameters such as learning rates.

## 4.2 One-Shot Formulation of the Knowledge Gradient using SAA

The acquisition functions mentioned above, such as EI and UCB, are *myopic*, that is, they do not take into account the effect of an observation on the model in future iterations. In contrast, *look-ahead* methods do. Our SAA approach enables a novel formulation of a class of look-ahead acquisition functions. For the purpose of this paper we focus on the Knowledge Gradient (KG) [27], but our methods extend to other look-ahead acquisition functions such as two-step EI [107].

KG quantifies the expected increase in the maximum of $f$ from obtaining the additional (random) observation data $\{\mathbf{x}, y_{\mathcal{D}}(\mathbf{x})\}$. KG often shows improved BO performance relative to simpler, myopic acquisition functions such as EI [90], but in its traditional form it is computationally expensive and

hard to implement, two challenges that we address in this work. Writing $\mathcal{D}_{\mathbf{x}} := \mathcal{D} \cup \{\mathbf{x}, \mathbf{y}_{\mathcal{D}}(\mathbf{x})\}$, we introduce a generalized variant of parallel KG (qKG) [106]:

$$\alpha_{\mathrm{KG}}(\mathbf{x}; \mathcal{D}) = \mathbb{E}\Big[\max_{x' \in \mathbb{X}} \mathbb{E}\big[g(f(x')) \,|\, \mathcal{D}_{\mathbf{x}}\big] \,|\, \mathcal{D}\Big] - \mu_{\mathcal{D}}^*, \tag{4}$$

with $\mu_{\mathcal{D}}^* := \max_{x \in \mathbb{X}} \mathbb{E}[g(f(x)) \,|\, \mathcal{D}]$. Equation (4) quantifies the expected increase in the maximum posterior mean of $g \circ f$ after gathering samples at $\mathbf{x}$. For simplicity, we only consider standard BO here, but extensions for multi-fidelity optimization [83, 110] are also available in BOTORCH.

Maximizing KG requires solving a nested optimization problem. The standard approach is to optimize the inner and outer problems separately, in an iterative fashion. The outer problem is handled using stochastic gradient ascent, with each gradient observation potentially being an average over multiple samples [106, 109]. For each sample $y_{\mathcal{D}}^i(\mathbf{x}) \sim y_{\mathcal{D}}(\mathbf{x})$, the inner problem $\max_{x_i \in \mathbb{X}} \mathbb{E}\big[f(x_i) \,|\, \mathcal{D}_{\mathbf{x}}^i\big]$ is solved numerically, either via another stochastic gradient ascent [109] or multi-start L-BFGS-B [26]. An unbiased stochastic gradient can then be computed by leveraging the envelope theorem. Alternatively, the inner problem can be discretized [106]. The computational expense of this nested optimization can be quite large; our main insight is that it may also be unnecessary.

We treat optimizing $\alpha_{\mathrm{KG}}(\mathbf{x}, \mathcal{D})$ in (4) as an entirely deterministic problem using SAA. Using the reparameterization trick, we express $y_{\mathcal{D}}(\mathbf{x}) = h_{\mathcal{D}}^y(\mathbf{x}, \epsilon)$ for some deterministic $h_{\mathcal{D}}$,[3] and draw $N$ fixed base samples $\{\epsilon^i\}_{i=1}^N$ for the outer expectation. The resulting MC approximation of KG is:

$$\hat{\alpha}_{\mathrm{KG},N}(\mathbf{x}; \mathcal{D}) = \frac{1}{N} \sum_{i=1}^N \max_{x_i \in \mathbb{X}} \mathbb{E}\big[g(f(x_i)) \,|\, \mathcal{D}_{\mathbf{x}}^i\big] - \mu^*. \tag{5}$$

**Theorem 2.** *Suppose conditions (i) and (ii) of Theorem 1 hold, and that (iii) $g(\cdot)$ is affine. If the base samples $\{\epsilon^i\}_{i \geq 1}$ are drawn i.i.d from $\mathcal{N}(0, 1)$, then (1) $\hat{\alpha}_{\mathrm{KG},N}^* \to \alpha_{\mathrm{KG}}^*$ a.s., (2) $\mathrm{dist}(\hat{\mathbf{x}}_{\mathrm{KG},N}^*, \mathcal{X}_{\mathrm{KG}}^*) \to 0$ a.s., and (3) $\forall \delta > 0, \exists K < \infty, \beta > 0$ s.t. $\mathbb{P}\big(\mathrm{dist}(\hat{\mathbf{x}}_{\mathrm{KG},N}^*, \mathcal{X}_{\mathrm{KG}}^*) > \delta\big) \leq K e^{-\beta N}$ for all $N \geq 1$.*

Theorem 2 also applies when using RQMC (Appendix D.3), in which case we again observe improved empirical convergence rates. In Appendix D.4, we prove that if $f_{\mathrm{true}}$ is drawn from the same GP prior as $f$ and $g(f) \equiv f$, then the MC-approximated KG *policy* (i.e., when (5) is maximized in each period to select measurements) is *asymptotically optimal* [27, 25, 83, 7], meaning that as the number of measurements tends to infinity, an optimal point $x^* \in \mathcal{X}_f^* := \arg\max_{x \in \mathbb{X}} f(x)$ is identified.

Conditional on the fixed base samples, (5) does not exhibit the nested structure used in the conventional formulation (which requires solving an optimization problem to get a noisy gradient estimate). Moving the maximization outside of the sample average yields the *equivalent* problem

$$\max_{\mathbf{x} \in \mathbb{X}} \hat{\alpha}_{\mathrm{KG},N}(\mathbf{x}, \mathcal{D}) \equiv \max_{\mathbf{x}, \mathbf{x}'} \frac{1}{N} \sum_{i=1}^N \mathbb{E}\big[g(f(x_i)) \,|\, \mathcal{D}_{\mathbf{x}}^i\big], \tag{6}$$

where $\mathbf{x}' := \{x^i\}_{i=1}^N \in \mathbb{X}^N$ represent "next stage" solutions, or "fantasy points." If $g$ is affine, the expectation in (6) admits an analytical expression. If not, we use another MC approximation of the form (2) with $N_I$ fixed inner based samples $E_I$.[4] The key difference from the envelope theorem approach [109] is that we do not solve the inner problem to completion for every fantasy point for every gradient step w.r.t. $\mathbf{x}$. Instead, we solve (6) jointly over $\mathbf{x}$ and the fantasy points $\mathbf{x}'$. The resulting optimization problem is of higher dimension, namely $(q + N)d$ instead of $qd$, but unlike the envelope theorem formulation it can be solved as a single problem, using methods for deterministic optimization. Consequently, we dub this KG variant the "One-Shot Knowledge Gradient" (OKG). The ability to auto-differentiate the involved quantities (including the samples $y_{\mathcal{D}}^i(\mathbf{x})$ and $\xi_{\mathcal{D}_{\mathbf{x}}}^i(\mathbf{x})$ through the posterior updates) w.r.t. $\mathbf{x}$ and $\mathbf{x}'$ allows BOTORCH to solve this problem effectively. The main limitation of OKG is the linear growth of the dimension of the optimization problem in $N$, which can be challenging to solve - however, in practical settings, we observe good performance for moderate $N$. We provide a simplified implementation of OKG in the following section.

## 5 Programmable Bayesian Optimization with BOTORCH

SAA provides an efficient and robust approach to optimizing MC acquisition functions through the use of deterministic gradient-based optimization. In this section, we introduce BOTORCH, a

complementary differentiable programming framework for Bayesian optimization research. Following the conceptual framework outlined in Figure 1, BOTORCH provides modular abstractions for representing and implementing sophisticated BO procedures. Operations are implemented as PyTorch modules that are highly parallelizable on modern hardware and end-to-end differentiable, which allows for efficient optimization of acquisition functions. Since the chain of evaluations on the sample level does not make any assumptions about the form of the posterior, BOTORCH's primitives can be directly used with any model from which re-parameterized posterior samples can be drawn, including probabilistic programs [99, 8], Bayesian neural networks [71, 87, 61, 41], and more general types of GPs [19]. In this paper, we focus on an efficient and scalable implementation of GPs, GPyTorch [29].

To illustrate the core components of BOTORCH, we demonstrate how both known and novel acquisition functions can readily be implemented. For the purposes of exposition, we show a set of simplified implementations here; details and additional examples are given in Appendices G and H.

## 5.1 Composing BOTORCH Modules for Multi-Objective Optimization

In our first example, we consider $q$ParEGO [20], a variant of ParEGO [54], a method for multi-objective optimziation.

```
1  weights = torch.distributions.Dirichlet(torch.ones(num_objectives)).sample()
2  scalarized_objective = GenericMCObjective(
3    lambda Y: 0.05 * (weights * Y).sum(dim=-1) + (weights * Y).min(dim=-1).values
4  )
5  qParEGO = qExpectedImprovement(model=model, objective=scalarized_objective)
6  candidates, values = optimize_acqf(qParEGO, bounds=box_bounds, q=1)
```

Code Example 1: Multi-objective optimization via augmented Chebyshev scalarizations.

Code Example 1 implements the inner loop of $q$ParEGO. We begin by instantiating a `GenericMCObjective` module that defines an augmented Chebyshev scalarization. This is an instance of BOTORCH's abstract `MCObjective`, which applies a transformation $g(\cdot)$ to samples $\xi$ from a posterior in its `forward($\xi$)` pass. In line 5, we instantiate an `MCAcquisitionFunction` module, in this case, `qExpectedImprovement`, parallel EI. Acquisition functions combine a model and the objective into a single module that assigns a utility $\alpha(\mathbf{x})$ to a candidate set $\mathbf{x}$ in its `forward` pass. Models can be any PyTorch module implementing a probabilistic model conforming to BOTORCH's basic `Model` API. Finally, candidate points are selected by optimizing the acquisition function, through the use of the `optimize_acqf()` utility function, which finds the candidates $\mathbf{x}^* \in \arg\max_{\mathbf{x}} \alpha(\mathbf{x})$. Auto-differentiation makes it straightforward to use gradient-based optimization even for complex acquisition functions and objectives. Our SAA approach permits the use of deterministic higher-order optimization to efficiently and reliably find $\mathbf{x}^*$.

In [6] it is shown how performing operations on independently modeled objectives yields better optimization performance when compared to modeling combined outcomes directly (e.g., for the case of calibrating the outputs of a simulator). `MCObjective` is a powerful abstraction that makes this straightforward. It can also be used to implement unknown (i.e. modeled) outcome constraints: BOTORCH implements a `ConstrainedMCObjective` to compute a feasibility-weighted objective using a sample-level differentiable relaxation of the feasibility [89, 28, 31, 58].

## 5.2 Implementing Parallel, Asynchronous Noisy Expected Improvement

Noisy EI (NEI) [58] is an extension of EI that is well-suited to highly noisy settings, such as A/B tests. Here, we describe a novel *full MC* formulation of NEI that extends the original one from [58] to joint parallel optimization and generic objectives. Letting $(\xi, \xi_{\mathrm{obs}}) \sim f_{\mathcal{D}}((\mathbf{x}, \mathbf{x}_{\mathrm{obs}}))$, our implementation avoids the need to characterize the (uncertain) best observed function value explicitly by averaging improvements on samples from the joint posterior over new and previously evaluated points:

$$\mathrm{qNEI}(\mathbf{x}; \mathcal{D}) = \mathbb{E}\big[\big(\max g(\xi) - \max g(\xi_{\mathrm{obs}})\big)_+ \mid \mathcal{D}\big]. \tag{7}$$

Code Example 2 provides an implementation of qNEI as formulated in (7). New MC acquisition functions are defined by extending an `MCAcquisitionFunction` base class and defining a `forward`

```
1  class qNoisyExpectedImprovement(MCAcquisitionFunction):
2    @concatenate_pending_points
3    def forward(self, X: Tensor) -> Tensor:
4      q = X.shape[-2]
5      X_full = torch.cat([X, match_shape(self.X_baseline, X)], dim=-2)
6      posterior = self.model.posterior(X_full)
7      samples = self.sampler(posterior)
8      obj = self.objective(samples)
9      obj_new = obj[...,:q].max(dim=-1).value
10     obj_prev = obj[...,q:].max(dim=-1).value
11     improvement = (obj_new - obj_prev).clamp_min(0)
12     return improvement.mean(dim=0).value
```

Code Example 2: Parallel Noisy EI

pass that compute the utility of a candidate x. In the constructor (not shown), the programmer sets X_baseline to an appropriate subset of the points at which the function was observed.

Like all MC acquisition functions, qNEI can be extended to support *asynchronous* candidate generation, in which a set $\tilde{\mathbf{x}}$ of *pending points* have been submitted for evaluation, but have not yet completed. This is done by concatenating pending points into x with the @concatenate_pending_points decorator. This allows us to compute the joint utility $\alpha(\mathbf{x} \cup \tilde{\mathbf{x}}; \Phi, \mathcal{D})$ of all points, pending and new, but optimize only with respect to the new x. This strategy also provides a natural way of generating parallel BO candidates using *sequential greedy* optimization [94]: We generate a single candidate, add it to the set of pending points, and proceed to the next. Due to submodularity of many common classes of acquisition functions (e.g., EI, UCB) [105], this approach can often yield better optimization performance compared to optimizing all candidate locations simultaneously (see Appendix F.2).

With the observed, pending, and candidate points (X_full) in hand, we use the Model's posterior() method to generate an object that represents the joint posterior across all points. The Posterior returned by posterior(x) represents $f_{\mathcal{D}}(\mathbf{x})$ (or $y_{\mathcal{D}}(\mathbf{x})$, if the observation_noise keyword argument is set to True), and may be be explicit (e.g. a multivariate normal in the case of GPs), or implicit (e.g. a container for a warmed-up MCMC chain). Next, samples are drawn from the posterior distribution p via a MCSampler, which employs the reparameterization trick [50, 85]. Given base samples $E \in \mathbb{R}^{N_s \times qm}$, a Posterior object produces $N_s$ samples $\xi_{\mathcal{D}} \in \mathbb{R}^{N_s \times q \times m}$ from the joint posterior. Its forward(p) pass draws samples $\xi_{\mathcal{D}}^i$ from p by automatically constructing base samples $E$. By default, BOTORCH uses RQMC via scrambled Sobol sequences [78]. Finally, these samples are mapped through the objective, and the expected improvement between the candidate point x and observed/pending points is computed by marginalizing the improvements on the sample level.

## 5.3 Look-ahead Bayesian Optimization with One-Shot KG

Code Example 3 shows a simplified OKG implementation, as discussed in Section 4.2.

```
1  class qKnowledgeGradient(OneShotAcquisitionFunction):
2    def forward(self, X: Tensor) -> Tensor:
3      X, X_f = torch.split(X, [X.size(-2) - self.N, self.N], dim=-2)
4      fant_model = self.model.fantasize(X=X, sampler=self.sampler, observation_noise=True)
5      inner_acqf = SimpleRegret(
6        fant_model, sampler=self.inner_sampler, objective=self.objective,
7      )
8      with settings.propagate_grads(True):
9        return inner_acqf(X_f).mean(dim=0).value
```

Code Example 3: Implementation of One-Shot KG

Here, the input X to forward is a concatenation of x and $N$ *fantasy points* $\mathbf{x}'$ (this setup ensures that OKG can be optimized using the same APIs as all other acquisition functions). After X is split into its components, we utilize the Model's fantasize(x, sampler) method that, given x and a MCSampler, constructs a batched set of $N$ *fantasy models* $\{f^i\}_{i=1}^N$ such that $f_{\mathcal{D}}^i(\mathbf{x}) \stackrel{d}{=} f_{\mathcal{D}_{\mathbf{x}}^i}(\mathbf{x}), \forall \mathbf{x} \in \mathbb{X}^q$, where $\mathcal{D}_{\mathbf{x}}^i := \mathcal{D} \cup \{\mathbf{x}, y_{\mathcal{D}}^i(\mathbf{x})\}$ is the original dataset augmented by a fantasy observation at x. The fantasy models provide a distribution over functions conditioned on future observations at x, which is used here to implement one-step look-ahead. SimpleRegret computes $\mathbb{E}\big[g(f(x_i)) \,|\, \mathcal{D}_{\mathbf{x}}^i\big]$ from (6) for each $i$ in batch mode. The propagate_grads context enables auto-differentiation through both the *generation* of the fantasy models and the *evaluation* of their respective posteriors at the points $\mathbf{x}'$.

# 6 Experiments

## 6.1 Exploiting Parallelism and Hardware Acceleration

BOTORCH utilizes inference and optimization methods designed to exploit parallelization via batched computation, and integrates closely with GPyTorch [29]. These model have fast test-time (predictive) distributions and sampling. This is crucial for BO, where the same models are evaluated many times in order to optimize the acquisition function. GPyTorch makes use of structure-exploiting algebra and local interpolation for $\mathcal{O}(1)$ computations in querying the predictive distribution, and $\mathcal{O}(T)$ for drawing a posterior sample at $T$ points, compared to the standard $\mathcal{O}(n^2)$ and $\mathcal{O}(T^3 n^3)$ computations [82].

Figure 4 reports wall times for *batch evaluation* of `qExpectedImprovement` at multiple candidate sets $\{\mathbf{x}^i\}_{i=1}^b$ for different MC samples sizes $N$, on both CPU and GPU for a GPyTorch GP. We observe significant speedups from running on the GPU, with scaling essentially linear in the batch size $b$, except for very large $b$ and $N$. Figure 5 shows between 10–40X speedups when using fast predictive covariance estimates over standard posterior inference in the same setting. The speedups grow slower on the GPU, whose cores do not saturate as quickly as on the CPU when doing standard posterior inference (for additional details see Appendix B). Together, batch evaluation and fast predictive distributions enable efficient, parallelized acquisition function evaluation for a very large number (tens of thousands) of points. This scalability allows us to implement and exploit novel highly parallelized initialization and optimization techniques.

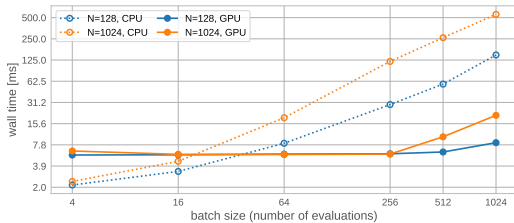

Figure 4: Wall times for batched evaluation of qEI

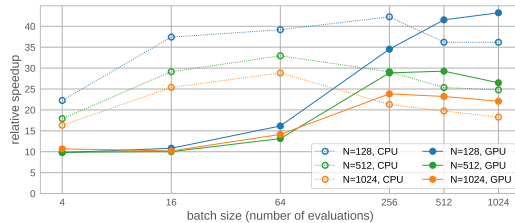

Figure 5: Fast predictive distributions speedups

## 6.2 Bayesian Optimization Performance Comparisons

We compare (i) the empirical performance of standard algorithms implemented in BOTORCH with those from other popular BO libraries, and (ii) our novel acquisition function, OKG, against other acquisition functions, both within BOTORCH and in other packages. We isolate three key frameworks—GPyOpt, Cornell MOE (*MOE EI*, *MOE KG*), and Dragonfly—because they are the most popular libraries with ongoing support[5] and are most closely related to BOTORCH in terms of state-of-the-art acquisition functions. GPyOpt uses an extension of EI with a local penalization heuristic (henceforth *GPyOpt LP-EI*) for parallel optimization [34]. For Dragonfly, we consider its default ensemble heuristic (henceforth *Dragonfly GP Bandit*) [49].

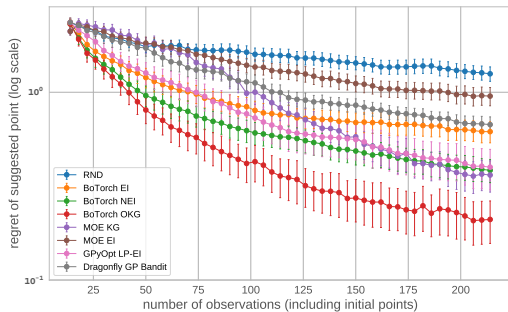

Figure 6: Hartmann ($d = 6$), noisy, best suggested

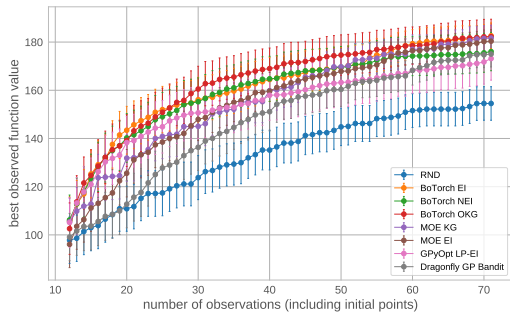

Figure 7: DQN tuning benchmark (Cartpole)

Our results provide three main takeaways. First, we find that BOTORCH's algorithms tend to achieve greater sample efficiency compared to those of other packages (all packages use their default models and settings). Second, we find that OKG often outperforms all other acquisition functions. Finally, OKG is more computationally scalable than MOE KG (the gold-standard implementation of KG), showing significant reductions in wall time (up to 6X, see Appendix C.2) while simultaneously achieving improved optimization performance (Figure 6).

**Synthetic Test Functions:** We consider BO for parallel optimization of $q = 4$ design points, on four noisy synthetic functions used in Wang et al. [100]: Branin, Rosenbrock, Ackley, and Hartmann. Figure 6 reports means and 95% confidence intervals over 100 trials for Hartmann; results for the other functions are qualitatively similar and are provided in Appendix C.1, together with details on the evaluation. Results for constrained BO using a differentiable relaxation of the feasibility indicator on the sample level are provided in Appendix C.3.

**Hyperparameter Optimization:** We illustrate the performance of BOTORCH on real-world applications, represented by three hyperparameter optimization (HPO) experiments: **(1)** Tuning 5 parameters of a deep Q-network (DQN) learning algorithm [66, 67] on the *Cartpole* task from OpenAI gym [12] and the default DQN agent implemented in Horizon [30], Figure 7; **(2)** Tuning 6 parameters of a neural network surrogate model for the UCI Adult data set [56] introduced by Falkner et al. [22], available as part of HPOlib2 [21], Figure 17 in Appendix C.4; **(3)** Tuning 3 parameters of the recently proposed *Stochastic Weight Averaging* (SWA) procedure of Izmailov et al. [40] on the VGG-16 [93] architecture for CIFAR-10, which achieves superior accuracy compared to previously reported results. A more detailed description of these experiments is given in Appendix C.4.

# 7 Discussion and Outlook

We presented a novel strategy for effectively optimizing MC acquisition functions using SAA, and established strong theoretical convergence guarantees (in fact, our RQMC convergence results are novel more generally, and of independent interest). Our proposed OKG method, an extension of this approach to "one-shot" optimization of look-ahead acquisition functions, constitutes a significant development of KG, improving scalability and allowing for generic composite objectives and outcome constraints. This approach can naturally be extended to multi-step and other look-ahead approaches [44].

We make these methodological and theoretical contributions available in our open-source library BOTORCH (https://botorch.org), a modern programming framework for BO that features a modular design and flexible API, our distinct SAA approach, and algorithms specifically designed to exploit modern computing paradigms such as parallelization and auto-differentiation. BOTORCH is particularly valuable in helping researchers to rapidly assemble novel BO techniques. Specifically, the basic MC acquisition function abstraction provides generic support for batch optimization, asynchronous evaluation, RQMC integration, and composite objectives (including outcome constraints).

Our empirical results show that besides increased flexibility, our advancements in both methodology and computational efficiency translate into significantly faster and more accurate closed-loop optimization performance on a range of standard problems. While other settings such as high-dimensional [47, 102, 59], multi-fidelity [83, 110], or multi-objective [54, 80, 20] BO, and non-MC acquisition functions such as Max-Value Entropy Search [101], are outside the scope of this paper, these approaches can readily be realized in BOTORCH and are included in the open-source software package. One can also naturally generalize BO procedures to incorporate neural architectures in BOTORCH using standard PyTorch models. In particular, deep kernel architectures [103], deep Gaussian processes [19, 88], and variational auto-encoders [33, 68] can easily be incorporated into BOTORCH's primitives, and can be used for more expressive kernels in high-dimensions.

In summary, BOTORCH provides the research community with a robust and extensible basis for implementing new ideas and algorithms in a modern computational paradigm, theoretically backed by our novel SAA convergence results.

**Broader Impact**

Bayesian optimization is a generic methodology for optimizing black-box functions, and therefore, by its very nature, not tied to any particular application domain. As mentioned earlier in the paper, Bayesian optimization has been used for various arguably good causes, including drug discovery or reducing the energy footprint of ML applications by reducing the computational cost of tuning hyperparameters. In the Appendix, we give an specific example for how our work can be applied in a public health context, namely to efficiently distribute survey locations for estimating malaria prevalence. BOTORCH as a tool specifically has been used in various applications, including transfer learning for neural networks [62], high-dimensional Bayesian optimization [59], drug discovery [10], sim-to-real transfer [69], trajectory optimization [42], and nano-material design [73]. However, there is nothing inherent to this work and Bayesian optimization as a field more broadly that would preclude it from being abused in some way, as is the case with any general methodology.

## Acknowledgments and Disclosure of Funding

We wish to thank Art Owen for insightful conversations on quasi-Monte-Carlo methods. We also express our appreciation to Peter Frazier and Javier Gonzalez for their helpful feedback on earlier versions of this paper.

Andrew Gordon Wilson is supported by NSF I-DISRE 193471, NIH R01 DA048764-01A1, NSF IIS-1910266, and NSF 1922658 NRT-HDR: FUTURE Foundations, Translation, and Responsibility for Data Science.

## Footnotes

[1]No implementation of ProBO is available at the time of this writing.

[2]Many utility functions $a$ are Lipschitz, including those representing (parallel) EI and UCB [104]. Lipschitzness is a sufficient condition, and convergence can also be shown in less restrictive settings (see Appendix D).

[3] For a GP, $h_{\mathcal{D}}^y(\mathbf{x}, \epsilon) = \mu_{\mathcal{D}}(\mathbf{x}) + L_{\mathcal{D}}^\sigma(\mathbf{x})\epsilon$, with $L_{\mathcal{D}}^\sigma(\mathbf{x})$ a root decomposition of $\Sigma_{\mathcal{D}}^\sigma(\mathbf{x}) := \Sigma_{\mathcal{D}}(\mathbf{x}, \mathbf{x}) + \Sigma^v(\mathbf{x})$.

[4] Convergence results can be established in the same way, and will require that $\min\{N, N_I\} \to \infty$.

[5]We were unable to install GPFlowOpt due to its incompatibility with current versions of GPFlow/TensorFlow.

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
