[Supplementary Material]

# Appendix to:

# BOTORCH: A Framework for Efficient Monte-Carlo Bayesian Optimization

## A  Brief Overview of Other Software Packages for BO

One of the earliest commonly-used packages is **Spearmint** [94], which implements a variety of modeling techniques such as MCMC hyperparameter sampling and input warping [95]. Spearmint also supports parallel optimization via fantasies, and constrained optimization with the expected improvement and predictive entropy search acquisition functions [31, 38]. Spearmint was among the first libraries to make BO easily accessible to the end user.

**GPyOpt** [98] builds on the popular GP regression framework GPy [35]. It supports a similar set of features as Spearmint, along with a local penalization-based approach for parallel optimization [34]. It also provides the ability to customize different components through an alternative, more modular API.

**Cornell-MOE** [106] implements the Knowledge Gradient (KG) acquisition function, which allows for parallel optimization, and includes recent advances such as large-scale models incorporating gradient evaluations [109] and multi-fidelity optimization [110]. Its core is implemented in C++, which provides performance benefits but renders it hard to modify and extend.

**RoBO** [52] implements a collection of models and acquisition functions, including Bayesian neural nets [96] and multi-fidelity optimization [51].

**Emukit** [97] is a Bayesian optimization and active learning toolkit with a collection of acquisition functions, including for parallel and multi-fidelity optimization. It does not provide specific abstractions for implementing new algorithms, but rather specifies a model API that allows it to be used with the other toolkit components.

The recent **Dragonfly** [49] library supports parallel optimization, multi-fidelity optimization [48], and high-dimensional optimization with additive kernels [47]. It takes an ensemble approach and aims to work out-of-the-box across a wide range of problems, a design choice that makes it relatively hard to extend.

## B  Parallelism and Hardware Acceleration

### B.1  Batch Evaluation

Batch evaluation, an important element of modern computing, enables automatic dispatch of independent operations across multiple computational resources (e.g. CPU and GPU cores) for parallelization and memory sharing. All BOTORCH components support batch evaluation, which makes it easy to write concise and highly efficient code in a platform-agnostic fashion. Batch evaluation enables fast queries of acquisition functions at a large number of candidate sets in parallel, facilitating novel initialization heuristics and optimization techniques.

Specifically, instead of sequentially evaluating an acquisition function at a number of candidate sets $\mathbf{x}_1, \ldots, \mathbf{x}_b$, where $\mathbf{x}_k \in \mathbb{R}^{q \times d}$ for each $k$, BOTORCH evaluates a batched tensor $\mathbf{x} \in \mathbb{R}^{b \times q \times d}$. Computation is automatically distributed so that, depending on the hardware used, speedups can be close to linear in the batch size $b$. Batch evaluation is also heavily used in computing MC acquisition functions, with the effect that significantly increasing the number of MC samples often has little impact on wall time. In Figure 4 we observe significant speedups from running on the GPU, with

Figure 8: Stochastic/deterministic opt. of EI on Hartmann6

Figure 9: Branin ($d = 2$)

Figure 10: Rosenbrock ($d = 3$)

Figure 11: Ackley ($d = 5$)

scaling essentially linear in the batch size, except for very large $b$ and $N$. The fixed cost due to communication overhead renders CPU evaluation faster for small batch and sample sizes.

# C   Additional Empirical Results

This section describes a number of empirical results that were omitted from the main paper due to space constraints.

## C.1   Synthetic Functions

Algorithms start from the same set of $2d + 2$ QMC sampled initial points for each trial, with $d$ the dimension of the design space. We evaluate based on the true noiseless function value at the "suggested point" (i.e., the point to be chosen *if BO were to end at this batch*). OKG, MOE KG, and NEI use "out-of-sample" suggestions (introduced as $\chi_n$ in Section D.4), while the others use "in-sample" suggestions [26].

All functions are evaluated with noise generated from a $\mathcal{N}(0, .25)$ distribution. Figures 9-11 give the results for all synthetic functions from Section 6. The results show that BOTORCH's NEI and OKG acquisition functions provide highly competitive performance in all cases.

## C.2   One-Shot KG Computational Scaling

Figure 12 shows the wall time for generating a set of $q = 8$ candidates as a function of the number of total data points $n$ for both standard (Cholesky-based) as well as scalable (Linear CG) posterior inference methods, on both CPU and GPU. While the GPU variants have a significant overhead for small models, they are significantly faster for larger models. Notably, our SAA based OKG is significantly faster than *MOE KG*, while at the same time achieving much better optimization performance (Figure 13).

Figure 12: KG wall times

Figure 13: Hartmann ($d = 6$), noisy, best suggested

Figure 14: Constrained Hartmann6, $f_2(x) = \|x\|_1 - 3$

Figure 15: Constrained Hartmann6, $f_1(x) = \|x\|_2 - 1$

### C.3 Constrained Bayesian Optimization

We present results for constrained BO on a synthetic function. We consider a multi-output function $f = (f_1, f_2)$ and the optimization problem:

$$\max_{x \in \mathbb{X}} \ f_1(x) \quad \text{s.t.} \quad f_2(x) \leq 0. \tag{8}$$

Both $f_1$ and $f_2$ are observed with $\mathcal{N}(0, 0.5^2)$ noise and we model the two components using independent GP models. A constraint-weighted composite objective is used in each of the BOTORCH acquisition functions EI, NEI, and OKG.

Results for the case of a Hartmann6 objective and two types of constraints are given in Figures 14-15 (we only show results for BOTORCH's algorithms, since the other packages do not natively support optimization subject to unknown constraints).

The regret values are computed using a feasibility-weighted objective, where "infeasible" is assigned an objective value of zero. For random search and EI, the suggested point is taken to be the best feasible noisily observed point, and for NEI and OKG, we use out-of-sample suggestions by optimizing the feasibility-weighted version of the posterior mean. The results displayed in Figure 15 are for the constrained Hartmann6 benchmark from [58]. Note, however, that the results here are not directly comparable to the figures in [58] because (1) we use feasibility-weighted objectives to compute regret and (2) they follow a different convention for suggested points. We emphasize that our contribution of outcome constraints for the case of KG has not been shown before in the literature.

### C.4 Hyperparameter Optimization Details

This section gives further detail on the experimental settings used in each of the hyperparameter optimization problems. As HPO typically involves long and resource intensive training jobs, it is standard to select the configuration with the best observed performance, rather than to evaluate a "suggested" configuration (we cannot perform noiseless function evaluations).

**DQN and Cartpole:** We consider the case of tuning a deep Q-network (DQN) learning algorithm [66, 67] on the *Cartpole* task from OpenAI gym [12] and the default DQN agent implemented in

Figure 16: DQN tuning benchmark (Cartpole)

Figure 17: NN surrogate model, best observed accuracy

Horizon [30]. Figure 16 shows the results of tuning five hyperparameters, *exploration parameter* ("epsilon"), the *target update rate*, the *discount factor*, the *learning rate*, and the *learning rate decay*. We allow for a maximum of 60 training episodes or 2000 training steps, whichever occurs first. To reduce noise, each "function evaluation" is taken to be an average of 10 independent training runs of DQN. Figure 16 presents the optimization performance of various acquisition functions from the different packages, using 15 rounds of parallel evaluations of size $q = 4$, over 100 trials. While in later iterations all algorithms achieve reasonable performance, BOTORCH OKG, EI, NEI, and GPyOpt LP-EI show faster learning early on.

**Neural Network Surrogate:** We consider the neural network surrogate model for the UCI Adult data set introduced by Falkner et al. [22], which is available as part of HPOlib2 [21]. We use a surrogate model to achieve a high level of precision in comparing the performance of the algorithms without incurring excessive computational training costs. This is a six-dimensional problem over network parameters (*number of layers*, *units per layer*) and training parameters (*initial learning rate*, *batch size*, *dropout*, *exponential decay factor for learning rate*). Figure 17 shows optimization performance in terms of best observed classification accuracy. Results are means and 95% confidence intervals computed from 200 trials with 75 iterations of size $q = 1$. All BOTORCH algorithms perform quite similarly here, with OKG doing slightly better in earlier iterations. Notably, they all achieve significantly better accuracy than all other algorithms.

**Stochastic Weight Averaging on CIFAR-10:** Our final example is for the recently proposed *Stochastic Weight Averaging* (SWA) procedure of Izmailov et al. [40], for which good hyperparameter settings are not fully understood. The setting is 300 epochs of training on the VGG-16 [93] architecture for CIFAR-10. We tune three SWA hyperparameters: *learning rate*, *update frequency*, and *starting iteration* using OKG. Izmailov et al. [40] report the mean and standard deviation of the test accuracy over three runs to be $93.64$ and $0.18$, respectively, which corresponds to a 95% confidence interval of $93.64 \pm 0.20$. We tune the problem to an average accuracy of $93.84 \pm 0.03$.

# D    Additional Theoretical Results and Omitted Proofs

## D.1    General SAA Results

Recall that we assume that $f(\mathbf{x}) \sim h(\mathbf{x}, \epsilon)$ for some $h : \mathbb{X} \times \mathbb{R}^s \to \mathbb{R}^{q \times m}$ and base random variable $\epsilon \in \mathbb{R}^s$ (c.f. Section 4 for an explicit expression for $h$ in case of a GP model). We write

$$A(\mathbf{x}, \epsilon) := a(g(h(\mathbf{x}, \epsilon))). \tag{9}$$

**Theorem 3** (Homem-de-Mello [39]). *Suppose that (i) $\mathbb{X}$ is a compact metric space, (ii) $\hat{\alpha}_N(\mathbf{x}) \xrightarrow{a.s.} \alpha(\mathbf{x})$ for all $\mathbf{x} \in \mathbb{X}^q$, and (iii) there exists an integrable function $\ell : \mathbb{R}^s \mapsto \mathbb{R}$ such that for almost every $\epsilon$ and all $\mathbf{x}, \mathbf{y} \in \mathbb{X}$,*

$$|A(\mathbf{x}, \epsilon) - A(\mathbf{y}, \epsilon)| \le \ell(\epsilon)\|\mathbf{x} - \mathbf{y}\|. \tag{10}$$

*Then $\hat{\alpha}_N^* \xrightarrow{a.s.} \alpha^*$ and $\mathrm{dist}(\hat{\mathbf{x}}_N^*, \mathcal{X}_f^*) \xrightarrow{a.s.} 0$.*

**Proposition 1.** *Suppose that (i) $\mathbb{X}$ is a compact metric space, (ii) $f$ is a GP with continuously differentiable prior mean and covariance functions, and (iii) $g(\cdot)$ and $a(\cdot, \Phi)$ are Lipschitz continuous. Then, condition (10) in Theorem 3 holds.*

The following proposition follows directly from Proposition 2.1, Theorem 2.3, and remarks on page 528 of [39].

**Proposition 2** (Homem-de-Mello [39])**.** *Suppose that, in addition to the conditions in Theorem 3, (i) the base samples $E = \{\epsilon^i\}_{i=1}^N$ are i.i.d., (ii) for all $\mathbf{x} \in \mathbb{X}^q$ the moment generating function $M_{\mathbf{x}}^A(t) := \mathbb{E}[e^{tA(\mathbf{x},\epsilon)}]$ of $A(\mathbf{x}, \epsilon)$ is finite in an open neighborhood of $t = 0$ and (iii) the moment generating function $M^\ell(t) := \mathbb{E}[e^{t\ell(\epsilon)}]$ is finite in an open neighborhood of $t = 0$. Then, there exist $K < \infty$ and $\beta > 0$ such that $\mathbb{P}(\text{dist}(\hat{\mathbf{x}}_N, \mathcal{X}_f^*)) \leq Ke^{-\beta N}$ for all $N \geq 1$.*

## D.2 Formal Statement of Theorem 1

**Theorem 1 (Formal Version).** *Suppose (i) $\mathbb{X}$ is compact, (ii) $f$ has a GP prior with continuously differentiable mean and covariance functions, and (iii) $g(\cdot)$ and $a(\cdot, \Phi)$ are Lipschitz continuous. If the base samples $\{\epsilon^i\}_{i=1}^N$ are drawn i.i.d. from $\mathcal{N}(0,1)$, then*

*(1) $\hat{\alpha}_N^* \to \alpha^*$ a.s., and*

*(2) $\text{dist}(\hat{\mathbf{x}}_N^*, \mathcal{X}^*) \to 0$ a.s.*

*If, in addition, (iii) for all $\mathbf{x} \in \mathbb{X}^q$ the moment generating function $M_{\mathbf{x}}^A(t) := \mathbb{E}[e^{tA(\mathbf{x},\epsilon)}]$ of $A(\mathbf{x}, \epsilon)$ is finite in an open neighborhood of $t = 0$ and (iv) the moment generating function $M^\ell(t) := \mathbb{E}[e^{t\ell(\epsilon)}]$ is finite in an open neighborhood of $t = 0$, then*

*(3) $\forall \delta > 0, \exists K < \infty, \beta > 0$ s.t. $\mathbb{P}\big(\text{dist}(\hat{\mathbf{x}}_N^*, \mathcal{X}^*) > \delta\big) \leq Ke^{-\beta N}$ for all $N \geq 1$.*

## D.3 Randomized Quasi-Monte Carlo Sampling for Sample Average Approximation

In order to use randomized QMC methods with SAA for MC acquisition function, the base samples $E = \{\epsilon^i\}$ will need to be generated via RQMC. For the case of Normal base samples, this can be achieved in various ways, e.g. by using inverse CDF methods or a suitable Box-Muller transform of samples $\epsilon^i \in [0,1]^s$ (both approaches are implemented in BOTORCH). In the language of Section 4, such a transform will become part of the base sample transform $\epsilon \mapsto h(\mathbf{x}, \epsilon)$ for any fixed $\mathbf{x}$.

For the purpose of this paper, we consider scrambled $(t, d)$-sequences as discussed by Owen [77], which are a particular class of RQMC method (BOTORCH uses PyTorch's implementation of scrambled Sobol sequences, which are $(t, d)$-nets in base 2). Using recent theoretical advances from Owen and Rudolf [79], it is possible to generalize the convergence results from Theorems 1 and 2 to the RQMC setting (to our knowledge, this is the first practical application of these theoretical results).

Let $(N_i)_{i \geq 1}$ be a sequence with $N_i \in \mathbb{N}$ s.t. $N_i \to \infty$ as $i \to \infty$. Then we have the following (see Appendix D.5 for the proofs):

**Theorem 1(q).** *In the setting of Theorem 1, let $\{\epsilon^i\}$ be samples from a $(t, d)$-sequence in base $b$ with gain coefficients no larger than $\Gamma < \infty$, randomized using a nested uniform scramble as in [77]. Then, the conclusions of Theorem 1 still hold. In particular,*

*(1) $\hat{\alpha}_{N_i}^* \to \alpha^*$ a.s. as $i \to \infty$,*

*(2) $\text{dist}(\hat{\mathbf{x}}_{N_i}^*, \mathcal{X}^*) \to 0$ a.s. as $i \to \infty$,*

*(3) $\forall \delta > 0, \exists K < \infty, \beta > 0$ s.t. $\mathbb{P}\big(\text{dist}(\hat{\mathbf{x}}_{N_i}^*, \mathcal{X}^*) > \delta\big) \leq Ke^{-\beta N_i}$ for all $i \geq 1$.*

**Theorem 2(q).** *In the setting of Theorem 2, let $\{\epsilon^i\}$ be samples from $(t, d)$-sequence in base $b$, with gain coefficients no larger than $\Gamma < \infty$, randomized using a nested uniform scramble as in [77]. Then,*

*(1) $\hat{\alpha}_{\text{KG},N_i}^* \xrightarrow{a.s.} \alpha_{\text{KG}}^*$ as $i \to \infty$,*

*(2) $\text{dist}(\hat{\mathbf{x}}_{\text{KG},N_i}^*, \mathcal{X}_{\text{KG}}^*) \xrightarrow{a.s.} 0$ as $i \to \infty$.*

Theorem 2(q) as stated does not provide a rate on the convergence of the optimizer. We believe that such result is achievable, but leave it to future work.

Note that while the above results hold for any sequence $(N_i)_i$ with $N_i \to \infty$, in practice the RQMC integration error can be minimized by using sample sizes that exploit intrinsic symmetry of the $(t, d)$-sequences. Specifically, for integers $b \geq 2$ and $M \geq 1$, let

$$\mathcal{N} := \{mb^k \mid m \in \{1, \dots, M\}, k \in \mathbb{N}_+\}. \tag{11}$$

In practice, we chose the MC sample size $N$ from the unique elements of $\mathcal{N}$.

## D.4  Asymptotic Optimality of OKG

Consider the case where $f_{\text{true}}$ is drawn from a GP prior with $f \overset{d}{=} f_{\text{true}}$, and that $g(f) \equiv f$. The KG *policy* (i.e., when used to select sequential measurements in a dynamic setting) is known to be *asymptotically optimal* [27, 25, 83, 7], meaning that as the number of measurements tends to infinity, an optimal point $x^* \in \mathcal{X}_f^* := \arg\max_{x \in \mathbb{X}} f(x)$ is identified. Although it does not necessarily signify good finite sample performance, this is considered a useful property for acquisition functions [25]. In this section, we state two results showing that OKG also possesses this property, providing further theoretical justification for the MC approach taken by BOTORCH.

Let $\mathcal{D}_0$ be the initial data and $\mathcal{D}_n$ for $n \geq 1$ be the data generated by taking measurements according to OKG using $N_n$ MC samples in iteration $n$, i.e., $\mathbf{x}_{n+1} \in \arg\max_{\mathbf{x} \in \mathbb{X}^q} \hat{\alpha}_{\text{KG},N_n}(\mathbf{x}; \mathcal{D}_n)$ for all $n$, and let $\chi_n \in \arg\max_{x \in \mathbb{X}} \mathbb{E}[f(x) \mid \mathcal{D}_n]$. Then we can show the following:

**Theorem 4.** *Suppose conditions (i) and (ii) of Theorem 1 and (iii) of Theorem 2 are satisfied. In addition, suppose that* $\limsup_n N_n = \infty$. *Then,* $f(\chi_n) \to f(x^*)$ *a.s. and in* $L^1$.

Theorem 4 shows that OKG is asymptotically optimal if the number of fantasies $N_n$ grows asymptotically with $n$ (this assumes we have an analytic expression for the inner expectation. If not, a similar condition must be imposed on the number of inner MC samples). In the special case of finite $\mathbb{X}$, we can quantify the sample sizes $\{N_n\}$ that ensure asymptotic optimality of OKG:

**Theorem 5.** *Along with conditions (i) and (ii) of Theorem 1, suppose that* $|\mathbb{X}| < \infty$ *and* $q = 1$. *Then, if for some* $\delta > 0$, $N_n \geq A_n^{-1} \log(K_n/\delta)$ *a.s., where* $A_n$ *and* $K_n$ *are a.s. finite and depend on* $\mathcal{D}_n$ *(these quantities can be computed), we have* $f(\chi_n) \to \max_{x \in \mathbb{X}} f(x)$ *a.s..*

## D.5  Proofs

In the following, we will denote by $\mu_{\mathcal{D}}(x) := \mathbb{E}[f(x) \mid \mathcal{D}]$ and $K_{\mathcal{D}}(x, y) := \mathbb{E}[(f(x) - \mathbb{E}[f(x)])(f(y) - \mathbb{E}[f(y)])^T \mid \mathcal{D}]$ the posterior mean and covariance functions of $f$ conditioned on data $\mathcal{D}$, respectively. Under some abuse of notation, we will use $\mu_{\mathcal{D}}(\mathbf{x})$ and $K_{\mathcal{D}}(\mathbf{x}, \mathbf{y})$ to denote multi point (vector / matrix)-valued variants of $\mu_{\mathcal{D}}$ and $K_{\mathcal{D}}$, respectively. If $f$ has a GP prior, then the posterior mean and covariance $\mu_{\mathcal{D}}(\mathbf{x})$ and $K_{\mathcal{D}}(\mathbf{x}, \mathbf{y})$ have well-known explicit expressions [84].

For notational simplicity and without loss of generality, we will focus on single-output GP case ($m = 1$) in this section. Indeed, in the multi-output case ($m > 1$), we have a GP $\hat{f}$ on $\mathbb{X} \times \mathbb{M}$ with $\mathbb{M} = \{1, \dots, m\}$, and covariance function $(x_1, i_1), (x_2, i_2) \mapsto \tilde{K}((x_1, i_1), (x_2, i_2))$. For $q = 1$ we then define $x \mapsto \tilde{f}(x) := [f(x, 0), \dots, f(x, m)]$, and then stack these for $q > 1$: $\mathbf{x} \mapsto [\tilde{f}(\mathbf{x}_1)^T, \dots, \tilde{f}(\mathbf{x}_q)^T]^T$. Then the analysis in the proofs below can be done on $mq$-dimensional and $mq \times mq$-dimensional posterior mean and covariance matrices (instead of $q$ and $q \times q$ dimensional ones for $m = 1$). Differentiability assumptions are needed only to establish certain boundedness results (e.g. in the proof of Proposition 1), but $\mathbb{M}$ is finite, so we will require differentiability of $K((\cdot, i_1), (\cdot, i_2))$ for each $i_1$ and $i_2$. Assumptions on other quantities can be naturally extended (e.g. for Theorem 2 $g$ will need to be Lipschitz on $\mathbb{R}^{q \times m}$ rather than on $\mathbb{R}^q$, etc.).

***Proof of Proposition 1***. Without loss of generality, we may assume $m = 1$ (the multi-output GP case follows immediately from applying the result below to $q' = qm$ and re-arranging the output). For a GP, we have $h_{\mathcal{D}}(\mathbf{x}, \epsilon) = \mu_{\mathcal{D}}(\mathbf{x}) + L_{\mathcal{D}}(\mathbf{x})\epsilon$ with $\epsilon \sim \mathcal{N}(0, I_q)$, where $\mu_{\mathcal{D}}(\mathbf{x})$ is the posterior mean and $L_{\mathcal{D}}(\mathbf{x})$ is the Cholesky decomposition of the posterior covariance $M_{\mathcal{D}}(\mathbf{x})$. It is easy to verify from the classic GP inference equations [84] that if prior mean and covariance function are continuously differentiable, then so are posterior mean $\mu_{\mathcal{D}}(\cdot)$ and covariance $K_{\mathcal{D}}(\cdot)$. Since the Cholesky decomposition is also continuously differentiable [70], so is $L_{\mathcal{D}}(\cdot)$. As $\mathbb{X}$ is compact and $\mu_{\mathcal{D}}(\cdot)$ and $L_{\mathcal{D}}(\cdot)$ are continuously differentiable, their derivatives are bounded. It follows from

the mean value theorem that there exist $C_\mu, C_L < \infty$ s.t. $\|\mu_\mathcal{D}(\mathbf{x}) - \mu_\mathcal{D}(\mathbf{y})\| \leq C_\mu \|\mathbf{x} - \mathbf{y}\|$ and $\|(L_\mathcal{D}(\mathbf{x}) - L_\mathcal{D}(\mathbf{y}))\epsilon\| \leq C_L \|\epsilon\| \|\mathbf{x} - \mathbf{y}\|$. Thus,

$$\begin{aligned}
\|h_\mathcal{D}(\mathbf{x}, \epsilon) - h_\mathcal{D}(\mathbf{y}, \epsilon)\| &= \|\mu_\mathcal{D}(\mathbf{x}) - \mu(\mathbf{y}) + (L_\mathcal{D}(\mathbf{x}) - L_\mathcal{D}(\mathbf{y}))\epsilon\| \\
&\leq \|\mu_\mathcal{D}(\mathbf{x}) - \mu_\mathcal{D}(\mathbf{y})\| + \|(L_\mathcal{D}(\mathbf{x}) - L_\mathcal{D}(\mathbf{y}))\epsilon\| \\
&\leq \ell_h(\epsilon) \|\mathbf{x} - \mathbf{y}\|
\end{aligned}$$

where $\ell_h(\epsilon) := C_\mu + C_L \|\epsilon\|$. Since, by assumption, $g(\cdot)$ and $a(\cdot; \Phi)$ are Lipschitz (say with constants $L_a$ and $L_g$, respectively), it follows that $\|A(\mathbf{x}, \epsilon) - A(\mathbf{y}, \epsilon)\| \leq L_a L_g \ell_h(\epsilon) \|\mathbf{x} - \mathbf{y}\|$. It thus suffices to show that $\ell_h(\epsilon)$ is integrable. To see this, note that $|\ell_h(\epsilon)| \leq C_\mu + C_L C \sum_i |\epsilon_i|$ for some $C < \infty$ (equivalence of norms), and that $\epsilon_i \sim \mathcal{N}(0, 1)$ is integrable. $\qquad\square$

**Lemma 1.** *Suppose that (i) $f$ is a GP with continuously differentiable prior mean and covariance function, and (ii) that $a(\cdot, \Phi)$ and $g(\cdot)$ are Lipschitz. Then, for all $\mathbf{x} \in \mathbb{X}^q$ the moment generating functions $M_\mathbf{x}^A(t) := \mathbb{E}[e^{tA(\mathbf{x}, \epsilon)}]$ of $A(\mathbf{x}, \epsilon)$ and $M^\ell(t) := \mathbb{E}[e^{t\ell(\epsilon)}]$ are finite for all $t \in \mathbb{R}$.*

**Proof of Lemma 1.** Recall that $h_\mathcal{D}(\mathbf{x}, \epsilon) = \mu_\mathcal{D}(\mathbf{x}) + L_\mathcal{D}(\mathbf{x})\epsilon$ for the case of $f$ being a GP, where $\mu_\mathcal{D}(\mathbf{x})$ is the posterior mean and $L_\mathcal{D}(\mathbf{x})$ is the Cholesky decomposition of the posterior covariance $K_\mathcal{D}(\mathbf{x})$. Mirroring the argument from the proof of Proposition 1, it is clear that $A(\mathbf{x}, \epsilon)$ is Lipschitz in $\epsilon$ for each $\mathbf{x} \in \mathbb{X}^q$, say with constant $\tilde{C}_L$. Note that this implies that $\mathbb{E}[|A(\mathbf{x}, \epsilon)|] < \infty$ for all $\mathbf{x}$. We can now appeal to results pertaining to the concentration of Lipschitz functions of Gaussian random variables: the Tsirelson-Ibragimov-Sudakov inequality [11, Theorem 5.5] implies that

$$\log M_\mathbf{x}^A(t) \leq \frac{t^2 \tilde{C}_L^2}{2} + t \, \mathbb{E}[A(\mathbf{x}, \epsilon)]$$

for any $t \in \mathbb{R}$, which is clearly finite for all $t$ since $\mathbb{E}[A(\mathbf{x}, \epsilon)] \leq \mathbb{E}[|A(\mathbf{x}, \epsilon)|]$. From the proof of Proposition 1, we know that $A(\mathbf{x}, \epsilon)$ is $\ell(\epsilon)$-Lipschitz in $\mathbf{x}$, where $\ell(\epsilon)$ is itself Lipschitz in $\epsilon$. Hence, the concentration result in Theorem 5.5 of [11] applies again, and we are done. $\qquad\square$

**Proof of Theorem 1.** Under the stated assumptions, Lemma 1 ensures that condition (10) in Theorem 3 holds. Further, note that the argument about Lipschitzness of $A(\mathbf{x}, \epsilon)$ in $\epsilon$ in the proof of Lemma 1 implies that $\mathbb{E}[|A(\mathbf{x}, \epsilon)|] < \infty$ for all $\mathbf{x} \in \mathbb{X}^q$. Since the $\{\epsilon^i\}_{i=1}^N$ are i.i.d, the strong law of large numbers implies that $\hat{\alpha}_N(\mathbf{x}) \to \alpha(\mathbf{x})$ a.s. for all $x \in \mathbb{X}$. Claims (1) and (2) then follow by applying Theorem 3, and claim (3) follows by applying Proposition 2. $\qquad\square$

**Proof of Theorem 1(q).** Mirroring the proof of Theorem 1, we need to show that $\hat{\alpha}_{N_i}(\mathbf{x}) \to \alpha(\mathbf{x})$ a.s. as $i \to \infty$ for all $\mathbf{x} \in \mathbb{X}^q$. For any $\mathbf{x} \in \mathbb{X}^q$ and any $\epsilon_0 \in \mathbb{R}^q$, we have (by convexity and monotonicity of $|x| \mapsto |x|^2$ and the Lipschitz assumption on $a$ and $g$) that

$$\begin{aligned}
|A(\mathbf{x}, \epsilon)|^2 &= |A(\mathbf{x}, \epsilon_0) + A(\mathbf{x}, \epsilon) - A(\mathbf{x}, \epsilon_0)|^2 \\
&\leq |A(\mathbf{x}, \epsilon_0)|^2 + |A(\mathbf{x}, \epsilon) - A(\mathbf{x}, \epsilon_0)|^2 \\
&\leq |A(\mathbf{x}, \epsilon_0)|^2 + L_a^2 L_g^2 \|h_\mathcal{D}(\mathbf{x}, \epsilon) - h_\mathcal{D}(\mathbf{x}, \epsilon_0)\|^2
\end{aligned}$$

where $h_\mathcal{D}(\mathbf{x}, \epsilon) = \mu_\mathcal{D}(\mathbf{x}) + L_\mathcal{D}(\mathbf{x})\Phi^{-1}(\epsilon)$ with $\Phi^{-1}$ the inverse CDF of $\mathcal{N}(0, 1)$, applied elementwise to the vector $\epsilon$ of qMC samples. Now choose $\epsilon_0 = (0.5, \ldots, 0.5)$, then

$$|A(\mathbf{x}, \epsilon)|^2 \leq |a(g(0))|^2 + L_a^2 L_g^2 \|h_\mathcal{D}(\mathbf{x}, \epsilon)\|^2$$

Since the $\{\epsilon^i\}$ are generated by a nested uniform scramble, we know from Owen [77] that $\epsilon \sim U[0,1]^q$, and therefore $\Phi^{-1}(\epsilon) \sim \mathcal{N}(0, I_q)$. Since affine transformations of Gaussians remain Gaussian, we have that $\mathbb{E}\left[\|h_\mathcal{D}(\mathbf{x}, \epsilon)\|^2\right] < \infty$. This shows that $A(\mathbf{x}, \epsilon) \in L^2([0,1]^q)$. That $\hat{\alpha}_{N_i}(\mathbf{x}) \to 0$ a.s. as $i \to \infty$ for all $\mathbf{x} \in \mathbb{X}^q$ now follows from Owen and Rudolf [79, Theorem 3]. $\qquad\square$

**Lemma 2.** *If $f$ is a GP, then $f_{\mathcal{D}_\mathbf{x}}(x') = h(x', \mathbf{x}, \epsilon, \epsilon_I)$, where $\epsilon \sim \mathcal{N}(0, I_q)$ and $\epsilon_I \sim \mathcal{N}(0, 1)$ are independent and $h$ is linear in both $\epsilon$ and $\epsilon_I$.*

***Proof of Lemma 2.*** This essentially follows from the property of a GP that the covariance conditioned on a new observation $(x, y)$ is independent of $y$.[6] We can write $f_{\mathcal{D}_\mathbf{x}}(x') = \mu_{\mathcal{D}_\mathbf{x}}(x') + L^\sigma_{\mathcal{D}_\mathbf{x}}(x')\epsilon_I$ , where

$$\mu_{\mathcal{D}_\mathbf{x}}(x') := \mu_{\mathcal{D}}(x') + K_{\mathcal{D}}(x', \mathbf{x})K^\sigma_{\mathcal{D}}(\mathbf{x})^{-1}L^\sigma_{\mathcal{D}}(\mathbf{x})\epsilon,$$

$L^\sigma_{\mathcal{D}}(\mathbf{x})$ is the Cholesky decomposition of $K^\sigma_{\mathcal{D}}(\mathbf{x}) := K_{\mathcal{D}}(\mathbf{x}, \mathbf{x}) + \mathrm{diag}(\sigma^2(\mathbf{x}_1), \ldots, \sigma^2(\mathbf{x}_q))$, and $L^\sigma_{\mathcal{D}_\mathbf{x}}(x')$ is the Cholesky decomposition of

$$K_{\mathcal{D}_\mathbf{x}}(x', x') := K(x', x') - K_{\mathcal{D}}(x', \mathbf{x})K^\sigma_{\mathcal{D}}(\mathbf{x})^{-1}K_{\mathcal{D}}(\mathbf{x}, x').$$

Hence, we see that $f_{\mathcal{D}_\mathbf{x}}(x') = h(x', \mathbf{x}, \epsilon, \epsilon_I)$, with

$$h(x', \mathbf{x}, \epsilon, \epsilon_I) = \mu_{\mathcal{D}}(x') + K_{\mathcal{D}}(x', \mathbf{x})K^\sigma_{\mathcal{D}}(\mathbf{x})^{-1}L^\sigma_{\mathcal{D}}(\mathbf{x})\epsilon + L^\sigma_{\mathcal{D}_\mathbf{x}}(x')\epsilon_I, \qquad (12)$$

which completes the argument. □

**Theorem 6.** *Let $(a_n)_{n\geq 1}$ be a sequence of non-negative real numbers such that $a_n \to 0$. Suppose that (i) $\mathbb{X}$ is a compact metric space, (ii) $f$ is a GP with continuous sample paths and continuous variance function $x \mapsto \sigma^2(x)$, and (iii) $(\mathbf{x}_n)_{n\geq 1}$ is such that $\alpha^n_{\mathrm{KG}}(\mathbf{x}_n) > \sup_{\mathbf{x}\in\mathbb{X}^q} \alpha^n_{\mathrm{KG}}(\mathbf{x}) - a_n$ infinitely often almost surely. Then $\alpha^n_{\mathrm{KG}}(\mathbf{x}) \to 0$ a.s. for all $\mathbf{x} \in \mathbb{X}^q$.*

***Proof of Theorem 6.*** Bect et al. [7] provide a proof for the case $q = 1$. Following their exposition, one finds that the only thing that needs to be verified in order to generalize their results to $q > 1$ is that condition (c) in their Definition 3.18 holds also for the case $q > 1$. What follows is the multi-point analogue of step (f) in the proof of their Theorem 4.8, which establishes this.

Let $\mu : \mathbb{X} \to \mathbb{R}$ and $K : \mathbb{X} \times \mathbb{X} \to \mathbb{R}_+$ denote mean and covariance function of $f$. Let $Z_\mathbf{x} := f(\mathbf{x}) + \mathrm{diag}(\sigma(\mathbf{x}))$, where $\sigma(\mathbf{x}) := (\sigma(\mathbf{x}_1), \ldots, \sigma(\mathbf{x}_q))$, with $\epsilon \sim \mathcal{N}(0, I_d)$ independent of $f$. Moreover, let $x^* \in \arg\max \mu(x)$. Following the same argument as Bect et al. [7], we arrive at the intermediate conclusion that $\mathbb{E}[\max\{0, W_{\mathbf{x},y}\}] = 0$, where $W_{\mathbf{x},y} := \mathbb{E}[f(y) \,|\, Z_\mathbf{x}] - \mathbb{E}[f(x^*) \,|\, Z_\mathbf{x}]$. We need to show that this implies that $\max_{x\in\mathbb{X}} f(x) = m(x^*)$.

Under some abuse of notation we will use $\mu$ and $K$ also as the vector / matrix-valued mean / kernel function. Let $K^\sigma(\mathbf{x}) := K(\mathbf{x}, \mathbf{x}) + \mathrm{diag}(\sigma(\mathbf{x}))$ and observe that

$$W_{\mathbf{x},y} = \mu(y) - \mu(x^*) + \mathbb{1}_{\{C(\mathbf{x})\succ 0\}}(K(y, \mathbf{x}) - K(x^*, \mathbf{x}))K^\sigma(\mathbf{x})^{-1}(Z_\mathbf{x} - \mu(\mathbf{x})),$$

i.e., $W_{\mathbf{x},y}$ is Gaussian with $\mathrm{Var}(W_{\mathbf{x},y}) = V(\mathbf{x}, y, x^*)V(\mathbf{x}, y, x^*)^T$, where $V(\mathbf{x}, y, x^*) := (K(y, \mathbf{x}) - K(x^*, \mathbf{x}))K^\sigma(\mathbf{x})^{-1}$. Since $\mathbb{E}[\max\{0, W_{\mathbf{x},y}\}] = 0$, we must have that $\mathrm{Var}(W_{\mathbf{x},y}) = 0$. If $K^\sigma(\mathbf{x}) \succ 0$, this means that $(K(y, \mathbf{x}) - K(x^*, \mathbf{x})) = 0_q$. But if $K^\sigma(\mathbf{x}) \not\succ 0$, then $K(\mathbf{x}, \mathbf{x}) \not\succ 0$, which in turn implies that $K(y, \mathbf{x}) = K(x^*, \mathbf{x}) = 0_q$. This shows that $K(y, \mathbf{x}) = K(x^*, \mathbf{x})$ for all $y \in \mathbb{X}$ and all $\mathbf{x} \in \mathbb{X}$. In particular, $K(x, y) = K(x, x^*)$ for all $y \in \mathbb{X}$. Thus, $K(x, x) - K(x, y) = K(x, x^*) - K(x, x^*)$ for all $y, x \in \mathbb{X}$, and therefore $\mathrm{Var}(f(x) - f(y)) = K(x, x) - K(x, y) - K(y, x) + K(y, y) = 0$. As in [7] we can conclude that this means that the sample paths of $f - \mu$ are constant over $\mathbb{X}$, and therefore $\max_{x\in\mathbb{X}} f(x) = m(x^*)$. □

***Proof of Theorem 2.*** From Lemma 2 we have that $f_{\mathcal{D}_\mathbf{x}}(x') = h(x', \mathbf{x}, \epsilon, \epsilon_I)$ with $h$ as in (12). Without loss of generality, we can absorb $\epsilon_I$ into $\epsilon$ for the purposes of showing that condition (10) holds for the mapping $A_{\mathrm{KG}}(\mathbf{x}, \epsilon) := \max_{x'\in\mathbb{X}} \mathbb{E}[g(f(x')) \,|\, \mathcal{D}_\mathbf{x}]$. Since the affine (and thus, continuously differentiable) transformation $g$ preserves the necessary continuity and differentiability properties, we can follow the same argument as in the proof of Theorem 1 of [108]. In particular, using continuous differentiability of GP mean and covariance function, compactness of $\mathbb{X}$, and continuous differentiability of $g$, we can apply the envelope theorem in the same fashion. From this, it follows that for any $\epsilon \in \mathbb{R}^q$ and for each $1 \leq l \leq q, 1 \leq k \leq d$, the restriction of $\mathbf{x} \mapsto A_{\mathrm{KG}}(\mathbf{x}, \epsilon)$ to the $k, l$-th coordinate is absolutely continuous for all $\mathbf{x}$, thus the partial derivative $\partial_{\mathbf{x}_{lk}}A_{\mathrm{KG}}(\mathbf{x}, \epsilon)$ exists a.e. Further, for each $l$ there exist $\Lambda_l \in \mathbb{R}^q$ with $\|\Lambda_l\| < \infty$ s.t. $|\partial_{\mathbf{x}_{kl}}A_{\mathrm{KG}}(\mathbf{x}, \epsilon)| \leq \Lambda_l^T|\varepsilon|$ a.e. on $\mathbb{X}^q$ (here $|\cdot|$ denotes the element-wise absolute value of a vector). This uniform bound on the partial derivatives can be used to show that $A_{\mathrm{KG}}$ is $\ell(\epsilon)$-Lipschitz. Indeed, writing the difference $A_{\mathrm{KG}}(\mathbf{y}, \epsilon) - A_{\mathrm{KG}}(\mathbf{x}, \epsilon)$ as a

sum of differences in each of the $qd$ components of $\mathbf{x}$ and $\mathbf{y}$, respectively, using the triangle inequality, absolute continuity of the element-wise restrictions, and uniform bound on the partial derivatives, we have that

$$|A_{\mathrm{KG}}(\mathbf{y}, \epsilon) - A_{\mathrm{KG}}(\mathbf{x}, \epsilon)| \leq \sum_{k=1}^{q} \sum_{l=1}^{d} \Lambda_l^T |\epsilon| |\mathbf{y}_{kl} - \mathbf{x}_{kl}| \leq \max_{1 \leq l \leq d} \left\{ \Lambda_l^T |\epsilon| \right\} \|\mathbf{y} - \mathbf{x}\|_1$$

and so $\ell(\epsilon) = \max_l \{ \Lambda_l^T |\epsilon| \}$. Going back to viewing $\epsilon$ as a random variable, it is straightforward to verify that $\ell(\epsilon)$ is integrable. Indeed,

$$\mathbb{E}[|\ell(\epsilon)|] \leq \max_l \left\{ \sum_{k=1}^{q} \Lambda_{lk} \mathbb{E}[|\epsilon_k|] \right\} = \sqrt{2/\pi} \max_l \left\{ \|\Lambda_l\|_1 \right\}.$$

Since $g$ is assumed to be affine in (iii), we can apply Lemma 2 to see that $\mathbb{E}[g(f(x')) \,|\, \mathcal{D}_{\mathbf{x}}]$ is a GP. Therefore, $A_{\mathrm{KG}}(\mathbf{x}, \epsilon)$ represents the maximum of a GP and its moment generating function $\mathbb{E}[e^{t A_{\mathrm{KG}}(\mathbf{x}, \epsilon)}]$ is finite for all $t$ by Lemma 4. This implies finiteness of its absolute moments [65, Exercise 9.15] and we have that $\mathbb{E}[|A_{\mathrm{KG}}(\mathbf{x}, \epsilon)|] < \infty$ for all $\mathbf{x} \in \mathbb{X}$. Since the $\{\epsilon^i\}$ are i.i.d, the strong law of large numbers ensures that $\hat{\alpha}_{\mathrm{KG}, N}(\mathbf{x}) \to \alpha_{\mathrm{KG}}(\mathbf{x})$ a.s. Theorem 3 now applies to obtain (1) and (2).

Moreover, by the analysis above, it holds that

$$\ell(\epsilon) = \max_l \{ \Lambda_l^T |\epsilon| \} \leq q \max_l \|\Lambda_l^T\|_\infty \|\epsilon\|_\infty =: \ell'(\epsilon),$$

so $\ell'(\epsilon)$ is also a Lipschitz constant for $A_{\mathrm{KG}}(\cdot, \epsilon)$. Here, the absolute value version (the second result) of Lemma 4 applies, so we have that $\mathbb{E}[e^{t \ell'(\epsilon)}]$ is finite for all $t$. The conditions of Proposition 2 are now satisfied and we have the desired conclusion. $\qquad\square$

***Proof of Theorem 2(q).*** In the RQMC setting, we have by Owen [77] that $\epsilon \sim U[0,1]^q$. Therefore, we are now interested in examining $\tilde{A}_{\mathrm{KG}}(\mathbf{x}, \epsilon) := A_{\mathrm{KG}}(\mathbf{x}, \Phi^{-1}(\epsilon))$, since $\Phi^{-1}(\epsilon) \sim \mathcal{N}(0, I_q)$. Following the same analysis as in the proof of Theorem 2, we have Lipschitzness of $\tilde{A}_{\mathrm{KG}}(\cdot, \epsilon)$:

$$|\tilde{A}_{\mathrm{KG}}(\mathbf{y}, \epsilon) - \tilde{A}_{\mathrm{KG}}(\mathbf{x}, \epsilon)| \leq \ell(\Phi^{-1}(\epsilon)) \|\mathbf{y} - \mathbf{x}\|_1,$$

where $\ell(\cdot)$ is as defined in the proof of Theorem 2. As before, $\ell(\Phi^{-1}(\epsilon))$ is integrable. Like in the proof of Theorem 2, $\tilde{A}_{\mathrm{KG}}(\mathbf{x}, \epsilon)$ is the maximum of a GP and its moment generating function $\mathbb{E}[e^{t \tilde{A}_{\mathrm{KG}}(\mathbf{x}, \epsilon)}]$ is finite for all $t$ by Lemma 4, implying finiteness of its second moment: $\mathbb{E}[\tilde{A}_{\mathrm{KG}}(\mathbf{x}, \epsilon)^2] < \infty$ for all $\mathbf{x} \in \mathbb{X}$. Thus, that $\tilde{A}_{\mathrm{KG}}(\mathbf{x}, \epsilon) \in L^2([0,1]^q)$ and $\hat{\alpha}_{\mathrm{KG}, N_i}(\mathbf{x}) \to \alpha_{\mathrm{KG}}(\mathbf{x})$ a.s. as $i \to \infty$ for all $\mathbf{x} \in \mathbb{X}^q$ follows from Owen and Rudolf [79, Theorem 3]. Theorem 3 now allows us to conclude (1) and (2). $\qquad\square$

The following Lemma will be used to prove Theorem 4:

**Lemma 3.** *Consider a Gaussian Process $f$ on $\mathbb{X} \subset \mathbb{R}^d$ with covariance function $K(\cdot, \cdot): \mathbb{X} \times \mathbb{X} \to \mathbb{R}$. Suppose that (i) $\mathbb{X}$ is compact, and (ii) $K$ is continuously differentiable. Then $f$ has continuous sample paths.*

***Proof of Lemma 3.*** Since $K$ is continuously differentiable and $\mathbb{X}$ is compact, $K$ is Lipschitz on $\mathbb{X} \times \mathbb{X}$, i.e., $\exists L < \infty$ such that $|K(x, y) - K(x', y')| \leq L(\|x - x'\| + \|y - y'\|)$ for all $(x, y), (x', y') \in \mathbb{X} \times \mathbb{X}$. Thus

$$\begin{aligned}
\mathbb{E}|f(x) - f(x)|^2 &= K(x, x) - 2K(x, y) + K(y, y) \\
&\leq |K(x, x) - K(x, y)| + |K(y, y) - K(x, y)| \\
&\leq 2L\|x - y\|
\end{aligned}$$

Since $\mathbb{X}$ is compact, there exists $C := \max_{x, y \in \mathbb{X}} \|x - y\| < \infty$. With this it is easy to verify that there exist $C' < \infty$ and $\eta > 0$ such that $2L\|x - y\| < C' |\log \|x - y\||^{-(1+\eta)}$ for all $x, y \in \mathbb{X}$. Continuity of the sample paths then follows from Theorem 3.4.1 in [3]. $\qquad\square$

***Proof of Theorem 4***. From Lemma 3 we know that the GP has continuous sample paths. If $\mathbf{x}_{n+1} \in \arg\max_{\mathbf{x} \in \mathbb{X}^q} \hat{\alpha}^n_{\mathrm{KG}, N_n}(\mathbf{x})$ for all $n$, the almost sure convergence of $\hat{\mathbf{x}}^n_{\mathrm{KG}, N_n}$ to the set of optimizers of $\alpha^n_{\mathrm{KG}}$ from Theorem 2 together with continuity of $\alpha^n_{\mathrm{KG}}$ (established in the proof of Theorem 2) implies that for all $\delta > 0$ and each $n \geq 1$, $\exists N_n < \infty$ such that $\alpha^n_{\mathrm{KG}}(\mathbf{x}_{n+1}) > \sup_{\mathbf{x} \in \mathbb{X}^q} \alpha^n_{\mathrm{KG}}(\mathbf{x}) - \delta$. As $\limsup_n N_n = \infty$, $\exists (a_n)_{n \geq 1}$ with $a_n \to 0$ such that $\alpha^n_{\mathrm{KG}}(\mathbf{x}_{n+1}) > \sup_{\mathbf{x} \in \mathbb{X}^q} \alpha^n_{\mathrm{KG}}(\mathbf{x}) - a_n$ infinitely often. That $\alpha^n_{\mathrm{KG}}(\mathbf{x}) \to 0$ a.s. for all $\mathbf{x} \in \mathbb{X}^q$ then follows from Theorem 6. The convergence result for $f(\chi_n)$ then follows directly from Proposition 4.9 in [7]. □

**Lemma 4.** *Let $f$ be a mean zero GP defined on $\mathbb{X}$ such that $|f(x)| < \infty$ almost surely for each $x \in \mathbb{X}$. It holds that the moment generating functions of $\sup_{x \in \mathbb{X}} f(x)$ and $\sup_{x \in \mathbb{X}} |f(x)|$ are both finite, i.e.,*

$$\mathbb{E}\big[e^{t \sup_{x \in \mathbb{X}} f(x)}\big] < \infty \quad and \quad \mathbb{E}\big[e^{t \sup_{x \in \mathbb{X}} |f(x)|}\big] < \infty$$

*for any $t \in \mathbb{R}$.*

***Proof of Lemma 4***. Let $\|f\| := \sup_{x \in \mathbb{X}} f$. Since the sample paths of $f$ are almost surely finite, the Borell-TIS inequality [2, Theorem 2.1] states that $\mathbb{E}\|f\| < \infty$. We first consider $t > 0$ and begin by re-writing the expectation as

$$
\begin{aligned}
\mathbb{E}\big[e^{t\|f\|}\big] &= \int_0^\infty \mathbb{P}\big(e^{t\|f\|} > u\big)\, du \\
&\leq 1 + \int_1^\infty \mathbb{P}\big(e^{t\|f\|} > u\big)\, du \\
&= 1 + \int_1^\infty \mathbb{P}\big(\|f\| - \mathbb{E}\|f\| > t^{-1}\log u - \mathbb{E}\|f\|\big)\, du \\
&= 1 + t e^{t\mathbb{E}\|f\|} \int_{-\mathbb{E}\|f\|}^\infty \mathbb{P}\big(\|f\| - \mathbb{E}\|f\| > u\big) e^{tu}\, du \\
&\leq 1 + t e^{t\mathbb{E}\|f\|} \left[\int_{\min\{-\mathbb{E}\|f\|, 0\}}^0 + \int_0^\infty\right] \mathbb{P}\big(\|f\| - \mathbb{E}\|f\| > u\big) e^{tu}\, du \\
&\leq 1 + \big|\mathbb{E}\|f\|\big| t e^{t\mathbb{E}\|f\|} + t e^{t\mathbb{E}\|f\|} \int_0^\infty \mathbb{P}\big(\|f\| - \mathbb{E}\|f\| > u\big) e^{tu}\, du, \qquad (13)
\end{aligned}
$$

where a change of variables is performed in the third equality. Let $\sigma_{\mathbb{X}}^2 := \sup_{x \in \mathbb{X}} \mathbb{E}[f(x)^2]$. We can now use the Borell-TIS inequality to bound the tail probability in (13) by $2e^{-u^2/(2\sigma_{\mathbb{X}}^2)}$, obtaining:

$$\mathbb{E}\big[e^{t\|f\|}\big] \leq 1 + \big|\mathbb{E}\|f\|\big| t e^{t\mathbb{E}\|f\|} + t e^{t\mathbb{E}\|f\|} \int_0^\infty 2e^{-u^2/(2\sigma_{\mathbb{X}}^2) + tu}\, du < \infty.$$

Similarly, for $t < 0$, we have:

$$
\begin{aligned}
\mathbb{E}\big[e^{t\||f|\|}\big] &= \int_0^\infty \mathbb{P}\big(e^{t\||f|\|} > u\big)\, du \\
&\leq 1 + \int_1^\infty \mathbb{P}\big(e^{t\||f|\|} > u\big)\, du \\
&= 1 + \int_1^\infty \mathbb{P}\big(\||f|\| - \mathbb{E}\|f\| < t^{-1}\log u - \mathbb{E}\|f\|\big)\, du \\
&= 1 - t e^{t\mathbb{E}\|f\|} \int_{-\infty}^{-\mathbb{E}\|f\|} \mathbb{P}\big(\||f|\| - \mathbb{E}\|f\| < u\big) e^{tu}\, du \\
&\leq 1 - t e^{t\mathbb{E}\|f\|} \left[\int_0^{\max\{-\mathbb{E}\|f\|, 0\}} + \int_{-\infty}^0\right] \mathbb{P}\big(\||f|\| - \mathbb{E}\|f\| < u\big) e^{tu}\, du \\
&\leq 1 - \big|\mathbb{E}\|f\|\big| t e^{t\mathbb{E}\|f\|} - t e^{t\mathbb{E}\|f\|} \int_{-\infty}^0 \mathbb{P}\big(\||f|\| - \mathbb{E}\|f\| < u\big) e^{tu}\, du, \qquad (14)
\end{aligned}
$$

The same can be done for (14) to conclude that $\mathbb{E}\big[e^{t\|f\|}\big] < \infty$ for all $t$. For the case of $\mathbb{E}\big[e^{t\||f|\|}\big]$ and $t > 0$, we use a similar line of analysis as (13) along with the observation that

$$\mathbb{P}\big(\||f|\| - \mathbb{E}\|f\| > u\big) \leq 2\,\mathbb{P}\big(\|f\| - \mathbb{E}\|f\| > u\big).$$

For $t < 0$, the result is clear because $|||f||| \geq 0$. $\qquad\square$

***Proof of Theorem 5.*** Since we are in the case of finite $\mathbb{X}$, let $\mu_n$ and $\Sigma_n$ denote the posterior mean vector and covariance matrix of our GP after conditioning on $\mathcal{D}_n$. First, we give a brief outline of the argument. We know from previous work (Lemma A.6 of [25] or Lemma 3 of [83]) that given a posterior distribution parameterized by $\mu$ and $\Sigma$, if $\alpha_{\text{KG}}(x; \mu, \Sigma) = 0$ for all $x \in \mathbb{X}$, then an optimal design is identified:

$$\arg\max_{x \in \mathbb{X}} \mu(x) = \arg\max_{x \in \mathbb{X}} f(x)$$

almost surely. Thus, we can use the true KG values as a "potential function" to quantify how the OKG policy performs asymptotically, even though we are never using the KG acquisition function for selecting points. We emphasize that the data that induce $\{\mu_n\}_{n \geq 0}$ and $\{\Sigma_n\}_{n \geq 0}$ are collected using the OKG policy.

By a martingale convergence argument, there exists a limiting posterior distribution described by random variables $(\mu_\infty, \Sigma_\infty)$, i.e., $\mu_n \to \mu_\infty$ and $\Sigma_n \to \Sigma_\infty$ almost surely [25, Lemma A.5]. Let $A \subseteq \mathbb{X}$ be a subset of the feasible space. As was done in the proof of Theorem 4 of [25], we define the event:

$$H_A = \big\{ \alpha_{\text{KG}}(x; \mu_\infty, \Sigma_\infty) > 0, \, x \in A \big\} \cap \big\{ \alpha_{\text{KG}}(x; \mu_\infty, \Sigma_\infty) = 0, \, x \notin A \big\}. \tag{15}$$

Note that $H_A$, for all possible subsets $A$, partition the sample space. Consider some $A \neq \emptyset$. By Lemma A.7 of [25], if $\alpha_{\text{KG}}(x; \mu_\infty, \Sigma_\infty) > 0$, then $x$ is measured a finite number of times, meaning that there exists an almost surely finite random variable $M_0$ such that on iterations after $N_0$, OKG stops sampling from $A$. By the definition of $H_A$ in (15), there must exist another random iteration index $M_1 \geq M_0$ such that when $n \geq N_1$,

$$\min_{x \in \mathcal{A}} \alpha_{\text{KG}}(x; \mu_n, \Sigma_n) > \max_{x \notin \mathcal{A}} \alpha_{\text{KG}}(x; \mu_n, \Sigma_n),$$

implying that the exact KG policy must prefer points in $\mathcal{A}$ over all others after iteration $M_1$. This implies that

$$H_A \subseteq \Big\{ \arg\max_{x \in \mathbb{X}} \hat{\alpha}_{\text{KG},N_n}(x, \mu_n, \Sigma_n) \not\subseteq \arg\max_{x \in \mathbb{X}} \alpha_{\text{KG}}(x, \mu_n, \Sigma_n), \, \forall n \geq M_1 - 1 \Big\} =: E,$$

because if not, then there exists an iteration after $M_0$ where an element from $A$ is selected, which is a contradiction. As shown in the proof of Lemma 2, the next period posterior mean $\mathbb{E}\big[g(f(x')) \,|\, \mathcal{D}_\mathbf{x}\big]$ is a GP. Therefore, by Lemma 4, the moment generating function of $\max_{x' \in \mathbb{X}} \mathbb{E}\big[g(f(x')) \,|\, \mathcal{D}_\mathbf{x}\big]$ is finite. Theorem 2.6 of [39] establishes that our choice of $N_n$ guarantees

$$\mathbb{P}\Big[ \arg\max_{x \in \mathbb{X}} \hat{\alpha}_{\text{KG},N_n}(x, \mu_n, \Sigma_n) \not\subseteq \arg\max_{x \in \mathbb{X}} \alpha_{\text{KG}}(x, \mu_n, \Sigma_n) \,|\, \mathcal{F}_n \Big] \leq \delta,$$

from which it follows that

$$\sum_{n=0}^{\infty} \log \mathbb{P}\Big[ \arg\max_{x \in \mathbb{X}} \hat{\alpha}_{\text{KG},N_n}(x, \mu_n, \Sigma_n) \not\subseteq \arg\max_{x \in \mathbb{X}} \alpha_{\text{KG}}(x, \mu_n, \Sigma_n) \,|\, \mathcal{F}_n \Big] = -\infty.$$

After writing the probability of $E$ as an infinite product and performing some manipulation, we see that the above condition implies that the probability of event $E$ is zero, and we conclude that $\mathbb{P}(H_A) = 0$ for any nonempty $A$. Therefore, $\mathbb{P}(H_\emptyset) = 1$ and $\alpha_{\text{KG}}(x; \mu_\infty, \Sigma_\infty) = 0$ for all $x$ almost surely. $\qquad\square$

# E  Illustration of Sample Average Approximation

QMC methods have been used in other applications in machine learning, including variational inference [13] and evolutionary strategies [86], but rarely in BO. Letham et al. [58] use QMC in the context of a specific acquisition function. BOTORCH's abstractions make it straightforward (and mostly automatic) to use QMC integration with any acquisition function.

Using SAA, i.e., fixing the base samples $E = \{\epsilon^i\}$, introduces a consistent bias in the function approximation. While i.i.d. re-sampling in each evaluation ensures that $\hat{\alpha}_N(\mathbf{x}, \Phi, \mathcal{D})$ and $\hat{\alpha}_N(\mathbf{y}, \Phi, \mathcal{D})$ are conditionally independent given $(\mathbf{x}, \mathbf{y})$, this no longer holds when fixing the base samples.

Figure 18: MC and QMC acquisition functions, with and without re-drawing the base samples between evaluations. The model is a GP fit on 15 points randomly sampled from $\mathbb{X} = [0, 1]^6$ and evaluated on the (negative) Hartmann6 test function. The acquisition functions are evaluated along the slice $x(\lambda) = \lambda \mathbf{1}$.

Figure 18 illustrates this behavior for EI (we consider the simple case of $q = 1$ for which we have an analytic ground truth available). The top row shows the MC and QMC version, respectively, when re-drawing base samples for every evaluation. The solid lines correspond to a single realization, and the shaded region covers four standard deviations around the mean, estimated across 50 evaluations. It is evident that QMC sampling significantly reduces the variance of the estimate. The bottom row shows the same functions for 10 different realizations of fixed base samples. Each of these realizations is differentiable w.r.t. $x$ (and hence $\lambda$ in the slice parameterization). In expectation (over the base samples), this function coincides with the true function (the dashed black line). Conditional on the base sample draw, however, the estimate displays a consistent bias. The variance of this bias (across re-drawing the base samples) is much smaller for the QMC versions.

Figure 19: Performance for optimizing QMC-based EI. Solid lines: fixed base samples, optimized via L-BFGS-B. Dashed lines: re-sampling base samples, optimized via Adam (lr=0.025).

Even thought the function *values* may show noticeable bias, the bias of the *maximizer* (in $\mathbb{X}$) is typically very small. Figure 19 illustrates this behavior, showing empirical cdfs of the relative gap $1 - \alpha(\hat{x}_N^*)/\alpha(x^*)$ and the distance $\|x^* - \hat{x}_N^*\|_2$ over 250 optimization runs for different numbers of samples, where $x^*$ is the optimizer of the analytic function EI, and $\hat{x}_N^*$ is the optimizer of the QMC approximation. The quality of the solution of the deterministic problem is excellent even for relatively small sample sizes, and generally better than of the stochastic optimizer.

Figure 20 shows empirical mean and variance of the metrics from Figure 19 as a function of the number of MC samples $N$ on a log-log scale. The stochastic optimizer used is Adam with a learning rate of 0.025 using an exponential moving average convergence criterion with a maximum of 250 optimization steps. Both for the SAA and the stochastic version we use the same number of random restart initial conditions generated from the same initialization heuristic.

Empirical asymptotic convergence rates can be obtained as the slopes of the OLS fit (dashed lines), and are given in Table 1. It is quite remarkable that in order to achieve the same error as the MC approximation with 4096 samples, the QMC approximation only requires 64 samples. This holds true for the bias and variance of the (relativized) optimal value as well as for the distance from the true optimizer. That said, as we are in a BO setting, we are not necessarily interested in the estimation error $\hat{\alpha}_N^* - \alpha^*$ of the optimum, but primarily in how far $x_N^*$ is from the true optimizer $x^*$.

Figure 20: Bias and variance of optimizer $x_N^*$ and true EI value $EI(x_N^*)$ evaluated at the optimizer as a function of the number of (Q)MC samples for both SAA and stochastic optimzation ("re-sample").

| | MC | QMC | MC$^\dagger$ | QMC$^\dagger$ |
|---|---|---|---|---|
| $\mathbb{E}[1 - \hat{\alpha}_N^*/\alpha^*]$ | $-0.52$ | $-0.95$ | $-0.34$ | $-0.63$ |
| $\mathrm{Var}(1 - \hat{\alpha}_N^*/\alpha^*)$ | $-1.17$ | $-2.12$ | $-0.61$ | $-0.83$ |
| $\mathbb{E}[\|x_N^* - x^*\|_2]$ | $-1.06$ | $-1.94$ | $-0.65$ | $-1.07$ |
| $\mathrm{Var}(\|x_N^* - x^*\|_2)$ | $-2.25$ | $-4.15$ | $-1.18$ | $-1.34$ |

Table 1: Empirical asymptotic convergence rates for the setting in Figure 20 ($^\dagger$denotes re-sampling + optimization with Adam).

A somewhat subtle point is that whether better optimization of the acquisition function results in improved closed-loop BO performance depends on the acquisition function as well as the underlying problem. More exploitative acquisition functions, such as EI, tend to show worse performance for problems with high noise levels. In these settings, not solving the EI maximization exactly adds randomness and thus induces additional exploration, which can improve closed-loop performance. While a general discussion of this point is outside the scope of this paper, BOTORCH does provide a framework for optimizing acquisition functions well, so that these questions can be compartmentalized and acquisition function performance can be investigated independently from the quality of optimization.

Perhaps the most significant advantage of using deterministic optimization algorithms is that, unlike for algorithms such as SGD that require tuning the learning rate, the optimization procedure is essentially hyperparameter-free. Figure 8 shows the closed-loop optimization performance of qEI for both deterministic and stochastic optimization for different optimizers and learning rates. While some of the stochastic variants (e.g. ADAM with learning rate 0.01) achieve performance similar to the deterministic optimization, the type of optimizer and learning rate matters. In fact, the rank order of SGD and ADAM w.r.t. to the learning rate is reversed, illustrating that selecting the right hyperparameters for the optimizer is itself a non-trivial problem.

# F    Additional Implementation Details

## F.1    Batch Initialization for Multi-Start Optimization

For most acquisition functions, the optimization surface is highly non-convex, multi-modal, and (especially for "improvement-based" ones such as EI or KG) often flat (i.e. has zero gradient) in much of the domain $\mathbb{X}$. Therefore, optimizing the acquisition function is itself a challenging problem.

The simplest approach is to use zeroth-order optimizers that do not require gradient information, such as DIRECT or CMA-ES [45, 37]. These approaches are feasible for lower-dimensional problems, but do not scale to higher dimensions. Note that performing parallel optimization over $q$ candidates in a $d$-dimensional feature space means solving a $qd$-dimensional optimization problem.

A more scalable approach incorporates gradient information into the optimization. As described in Section 4, BOTORCH by default uses quasi-second order methods, such as L-BFGS-B. Because of the complex structure of the objective, the initial conditions for the algorithm are extremely important so as to avoid getting stuck in a potentially highly sub-optimal local optimum. To reduce this risk, one typically employs multi-start optimization (i.e. start the solver from multiple initial conditions and pick the best of the final solutions). To generate a good set of initial conditions, BOTORCH heavily exploits the fast batch evaluation discussed in the previous section. Specifically, BOTORCH by default uses $N_{\text{opt}}$ initialization candidates generated using the following heuristic:

1. Sample $\tilde{N}_0$ quasi-random $q$-tuples of points $\tilde{\mathbf{x}}_0 \in \mathbb{R}^{\tilde{N}_0 \times q \times d}$ from $\mathbb{X}^q$ using quasi-random Sobol sequences.
2. Batch-evaluate the acquisition function at these candidate sets: $\tilde{v} = \alpha(\tilde{\mathbf{x}}_0; \Phi, \mathcal{D})$.
3. Sample $N_0$ candidate sets $\mathbf{x} \in \mathbb{R}^{N_0 \times q \times d}$ according to the weight vector $p \propto \exp(\eta v)$, where $v = (\tilde{v} - \hat{\mu}(\tilde{v}))/\hat{\sigma}(\tilde{v})$ with $\hat{\mu}$ and $\hat{\sigma}$ the empirical mean and standard deviation, respectively, and $\eta > 0$ is a temperature parameter. Acquisition functions that are known to be flat in large parts of $\mathbb{X}^q$ are handled with additional care in order to avoid starting in locations with zero gradients.

Sampling initial conditions this way achieves an exploration/exploitation trade-off controlled by the magnitude of $\eta$. As $\eta \to 0$ we perform Sobol sampling, while $\eta \to \infty$ means the initialization is chosen in a purely greedy fashion. The latter is generally not advisable, since for large $\tilde{N}_0$ the highest-valued points are likely to all be clustered together, which would run counter to the goal of multi-start optimization. Fast batch evaluation allows evaluating a large number of samples ($\tilde{N}_0$ in the tens of thousands is feasible even for moderately sized models).

### F.2 Sequential Greedy Batch Optimization

The pending points approach discussed in Section 5 provides a natural way of generating parallel BO candidates using *sequential greedy* optimization, where candidates are chosen sequentially, while in each step conditioning on selected points and integrating over the uncertainty in their outcome (using MC integration). By using a full MC formulation, in which we jointly sample at new and pending points, we avoid constructing an individual "fantasy" model for each sampled outcome, a common (and costly) approach in the literature [94]. In practice, the sequential greedy approach often performs well, and may even outperform the joint optimization approach, since it involves a sequence of small, simpler optimization problems, rather than a larger and complex one that is harder to solve.

[105] provide a theoretical justification for why the sequential greedy approach works well with a class of acquisition functions that are submodular.

## G  Active Learning Example

Recall from Section 5 the negative integrated posterior variance (NIPV) [91, 17] of the model:

$$\text{NIPV}(\mathbf{x}) = - \int_{\mathbb{X}} \mathbb{E}\big[\text{Var}(f(x)\,|\,\mathcal{D}_{\mathbf{x}}) \,|\, \mathcal{D}\big]\, dx. \tag{16}$$

We can implement (16) using standard BOTORCH components, as shown in Code Example 4. Here `mc_points` is the set of points used for MC-approximating the integral. In the most basic case, one can use QMC samples drawn uniformly in $\mathbb{X}$. By allowing for arbitrary `mc_points`, we permit weighting regions of $\mathbb{X}$ using non-uniform sampling. Using `mc_points` as samples of the maximizer of the posterior, we recover the recently proposed Posterior Variance Reduction Search [74] for BO.

This acquisition function supports both parallel selection of points and asynchronous evaluation. Since MC integration requires evaluating the posterior variance at a large number of points, this acquisition function benefits significantly from the fast predictive variance computations in GPyTorch [82, 29].

```
1  class qNegativeIntegratedPosteriorVariance(AnalyticAcquisitionFunction):
2
3      @concatenate_pending_points
4      @t_batch_mode_transform()
5      def forward(self, X: Tensor) -> Tensor:
6          fant_model = self.model.fantasize(
7              X=X, sampler=self._dummy_sampler,
8              observation_noise=True
9          )
10         sz = [1] * len(X.shape[:-2]) + [-1, X.size(-1)]
11         mc_points = self.mc_points.view(*sz)
12         with settings.propagate_grads(True):
13             posterior = fant_model.posterior(mc_points)
14         ivar = posterior.variance.mean(dim=-2)
15         return -ivar.view(X.shape[:-2])
```

Code Example 4: Active Learning (NIPV)

To illustrate how NIPV may be used in combination with scalable probabilistic modeling, we examine the problem of efficient allocation of surveys across a geographic region. Inspired by Cutajar et al. [18], we utilize publicly-available data from The Malaria Atlas Project (2019) dataset, which includes the yearly mean parasite rate (along with standard errors) of *Plasmodium falciparum* at a $4.5\text{km}^2$ grid spatial resolution across Africa. In particular, we consider the following active learning problem: given a spatio-temporal probabilistic model fit to data from 2011-2016, which geographic locations in and around Nigeria should one sample in 2017 in order to minimize the model's error for 2017 across all of Nigeria?

We fit a heteroskedastic GP model to 2500 training points prior to 2017 (using a noise model that is itself a GP fit to the provided standard errors). We then select $q = 10$ sample locations for 2017 using the NIPV acquisition function, and make predictions across the entirety of Nigeria using this new data. Compared to using no 2017 data, we find that our new dataset reduces MSE by 16.7% on average (SEM = 0.96%) across 60 subsampled datasets. By contrast, sampling the new 2017 points at a regularly spaced grid results only in a 12.4% reduction in MSE (SEM = 0.99%). The mean relative improvement in MSE reduction from NIPV optimization is 21.8% (SEM = 6.64%). Figure 21 shows the NIPV-selected locations on top of the base model's estimated parasite rate and standard deviation.

Figure 21: Locations for 2017 samples from IPV minimization and the base grid. Observe how the NIPV samples cluster in higher variance areas.

Figure 22: Composite function optimization for $q = 1$  Figure 23: Composite function optimization for $q = 3$

# H    Additional Implementation Examples

## Comparing Implementation Complexity

Many of BOTORCH's benefits are qualitative, including the simplification and acceleration of implementing new acquisition functions. Quantifying this in a meaningful way is very challenging. Comparisons are often made in terms of Lines of Code (LoC) - while this metric is problematic when comparing across different design philosophies, non-congruent feature sets, or even programming languages, it does provides a general idea of the effort required for developing and implementing new methods.

MOE's KG involves thousands of LoC in C++ and python spread across a large number of files,[7] while our more efficient implementation is <30 LoC. Astudillo and Frazier [6] is a full paper in last year's installment of this conference,[8] whose composite function method we implement and significantly extend (e.g to support KG) in 7 LoC using BoTorch's abstractions. The original NEI implementation is >250 LoC, while the one from Code Example 2 is 14 LoC.

## H.1    Composite Objectives

We consider the Bayesian model calibration of a simulator with multiple outputs from Section 5.3 of Astudillo and Frazier [6]. In this case, the simulator from Bliznyuk et al. [9] models the concentrations of chemicals at 12 positions in a one-dimensional channel. Instead of modeling the overall loss function (which measures the deviation of the simulator outputs with a set of observations) directly, we follow Astudillo and Frazier [6] and model the underlying concentrations while utilizing a composite objective approach. A powerful aspect of BOTORCH's modular design is the ability to easily combine different approaches into one. For the composite function problem in this section this means that we can easily extend the work by Astudillo and Frazier [6] not only to use the Knowledge Gradient, but also to the "parallel BO" setting of jointly selecting $q > 1$ points. Figures 22 and 22 show results for this with $q = 1$ and $q = 3$, repspectively. The plots show log regret evaluated at the maximizer of the posterior mean averaged over 250 trials. While the performance of EI-CF is similar for $q = 1$ and $q = 3$, KG-CF reaches lower regret significantly faster for $q = 1$ compared to $q = 3$, suggesting that "looking ahead" is beneficial in this context.

## H.2    Generalized UCB

Code Example 5 presents a generalized version of parallel UCB from Wilson et al. [104] supporting pending candidates, generic objectives, and QMC sampling. If no `sampler` is specified, a default QMC sampler is used. Similarly, if no `objective` is specified, the identity objective is assumed.

## H.3    Full Code Examples

In this section we provide full implementations for the code examples. Specifically, we include parallel Noisy EI (Code Example 6), OKG (Code Example 7), and (negative) Integrated Posterior Variance (Code Example 8).

```
1 class qUpperConfidenceBound(MCAcquisitionFunction):
2
3     def __init__(
4         self,
5         model: Model,
6         beta: float,
7         sampler: Optional[MCSampler] = None,
8         objective: Optional[MCAcquisitionObjective] = None,
9         X_pending: Optional[Tensor] = None,
10    ) -> None:
11        super().__init__(model, sampler, objective, X_pending)
12        self.beta_prime = math.sqrt(beta * math.pi / 2)
13
14    @concatenate_pending_points
15    @t_batch_mode_transform()
16    def forward(self, X: Tensor) -> Tensor:
17        posterior = self.model.posterior(X)
18        samples = self.sampler(posterior)
19        obj = self.objective(samples)
20        mean = obj.mean(dim=0)
21        z = mean + self.beta_prime * (obj - mean).abs()
22        return z.max(dim=-1).values.mean(dim=0)
```

Code Example 5: Generalized Parallel UCB

```
1 class qNoisyExpectedImprovement(MCAcquisitionFunction):
2
3     def __init__(
4         self,
5         model: Model,
6         X_baseline: Tensor,
7         sampler: Optional[MCSampler] = None,
8         objective: Optional[MCAcquisitionObjective] = None,
9         X_pending: Optional[Tensor] = None,
10    ) -> None:
11        super().__init__(model, sampler, objective, X_pending)
12        self.register_buffer("X_baseline", X_baseline)
13
14    @concatenate_pending_points
15    @t_batch_mode_transform()
16    def forward(self, X: Tensor) -> Tensor:
17        q = X.shape[-2]
18        X_bl = match_shape(self.X_baseline, X)
19        X_full = torch.cat([X, X_bl], dim=-2)
20        posterior = self.model.posterior(X_full)
21        samples = self.sampler(posterior)
22        obj = self.objective(samples)
23        obj_n = obj[...,:q].max(dim=-1).values
24        obj_p = obj[...,q:].max(dim=-1).values
25        return (obj_n - obj_p).clamp_min(0).mean(dim=0)
```

Code Example 6: Parallel Noisy EI (full)

```
1  class qKnowledgeGradient(MCAcquisitionFunction):
2
3      def __init__(
4          self,
5          model: Model,
6          sampler: MCSampler,
7          objective: Optional[Objective] = None,
8          inner_sampler: Optional[MCSampler] = None,
9          X_pending: Optional[Tensor] = None,
10     ) -> None:
11         super().__init__(model, sampler, objective, X_pending)
12         self.inner_sampler = inner_sampler
13
14     def forward(self, X: Tensor) -> Tensor:
15         splits = [X.size(-2) - self.Nf, self.N_f]
16         X, X_fantasies = torch.split(X, splits, dim=-2)
17         # [...] some re-shaping for batch evaluation purposes
18         if self.X_pending is not None:
19             X_p = match_shape(self.X_pending, X)
20             X = torch.cat([X, X_p], dim=-2)
21         fmodel = self.model.fantasize(
22             X=X,
23             sampler=self.sampler,
24             observation_noise=True,
25         )
26         obj = self.objective
27         if isinstance(obj, MCAcquisitionObjective):
28             inner_acqf = SimpleRegret(
29                 fmodel, sample=self.inner_sampler, objective=obj,
30             )
31         else:
32             inner_acqf = PosteriorMean(fmodel, objective=obj)
33         with settings.propagate_grads(True):
34             values = inner_acqf(X_fantasies)
35         return values.mean(dim=0)
```

Code Example 7: One-Shot Knowledge Gradient (full)

```python
1  class qNegIntegratedPosteriorVariance(AnalyticAcquisitionFunction):
2      def __init__(
3          self,
4          model: Model,
5          mc_points: Tensor,
6          X_pending: Optional[Tensor] = None,
7      ) -> None:
8          super().__init__(model=model)
9          self._dummy_sampler = IIDNormalSampler(1)
10         self.X_pending = X_pending
11         self.register_buffer("mc_points", mc_points)
12
13     @concatenate_pending_points
14     @t_batch_mode_transform()
15     def forward(self, X: Tensor) -> Tensor:
16         fant_model = self.model.fantasize(
17             X=X,
18             sampler=self._dummy_sampler,
19             observation_noise=True,
20         )
21         batch_ones = [1] * len(X.shape[:-2])
22         mc_points = self.mc_points.view(*batch_ones, -1, X.size(-1))
23         with settings.propagate_grads(True):
24             posterior = fant_model.posterior(mc_points)
25         ivar = posterior.variance.mean(dim=-2)
26         return -ivar.view(X.shape[:-2])
```

Code Example 8: Active Learning (full)

## Footnotes

[6]In some cases we may consider constructing a heteroskedastic noise model that results in the function $\sigma^2(\mathbf{x})$ changing depending on observations $y$, in which case this argument does not hold true anymore. We will not consider this case further here.

[7] https://github.com/wujian16/Cornell-MOE

[8] Code available at https://github.com/RaulAstudillo06/BOCF