[Reviews · NeurIPS 2020]

Review 1

Summary and Contributions: EDIT: I've read the rebuttal. ----------------------------------------------------------------------------- This work presents a Bayesian optimization library -- BOTorch, which has advantages over other existing BO libraries: * A novel approach to optimize MC acquisition functions using fixed sample averages. This leads to much faster and easier implementation of second-order methods and look-ahead BO methods (e.g., Knowledge gradient). * Novel general convergence results for sample-average approximation to acquisition functions via randomized quasi-MC.

Strengths: * This BO library is largely distinct from existing ones due to its approach to MC acquisition functions --- a fixed sample average is used and optimized throughout the whole process. This allows approximate 2nd-order methods such as L-BFGS to be applied stably, which is not the case if samples are repeatedly drawn to evaluate the expectation, as done in stochastic gradient methods. * A wide range of experiments are conducted to demonstrate the performance improvement and the results are convincing compared to multiple baselines (implemented in the same and other libraries). * Using this approach, BOTorch provides a fast implementation of the Knowledge gradient method: one-shot KG, which gets rid of inner optimization loops. This seems to be a significant contribution to the KG algorithm and has potential applications in other look-ahead BO methods.

Weaknesses: * As a software paper, probably due to space limits, section 5 is too brief for readers to understand the main concepts in the library, and most features described there aren't illustrated with code. As a result, it is very difficult to understand the code segments provided in the text. * L179: BOTORCH provides a @concatenate_pending_points decorator to 180 add this functionality to any MC acquisition function. I don't think the mechanism of parallel and asynchronous BO in BOTorch is described somewhere in the paper.

Correctness: Overall the paper is technically sound, although I did not check the proof of convergence results (the proof is directly adapted from Homem-de-Mello (2008).

Clarity: The methodology is clear. The clarity can be improved if the library feature description is paired with code examples.

Relation to Prior Work: The comparison to GPFlowOpt can be more properly executed. * L50: GPFlowOpt inherits support for auto-differentiation and hardware acceleration from TensorFlow [via GPFlow, 60], but unlike BOTORCH, it does not use algorithms designed to specifically exploit this potential. Can you be more specific about what the potential you are referring to here?

Reproducibility: Yes

Additional Feedback: * It is not required but, since this is a known open-source library from a big company, should the broader impact section focus more on how it will be deployed in products and the potential negative impacts?


Review 2

Summary and Contributions: The authors introduce BoTorch, a framework for BO. BoTorch is a collection of useful techniques that make BO efficient, including MC acquisition functions, SAA optimization, auto-differentiation, etc. BoTorch facilitates the specification of new acquisition functions. Theoretical convergence results are provided for the SAA BO approach and for the One-shot KG formulation.

Strengths: This paper is highly relevant for the NeurIPS community and tackles the important problem of Bayesian Optimization (BO). The claims are sound, significant and novel. From a technical perspective the authors put a lot of effort in implementing a well-thought framework for BO that is realised open-source. This is by itself a great technical contribution that will greatly benefit the BO community. From the theoretical perspective the idea of using SAA and the theoretical garantees are of important value and given the evidence provided in the paper score well compared to stochastic approaches. The real added value of SAA is transforming the optimization into a deterministic optimization setting where standard optimization techniques can be used. The One-shot formulation of KG and its theoretical result is also relevant to the BO community given the importance of the KG method. The novel techniques have been used in a series of satisfying experiments.

Weaknesses: Perhaps my main remark is that the paper could have been structured differently. The format of the paper is unconventional, i.e., the theoretical results and the experiments are mixed. I'd suggest to use a more conventional writing style to benefit the reader. The background section can be extended to make the reader more confortable with some of the concept that are heavily used in the rest of the text, e.g., fantasies, KG. While some of these are briefly introduced later in the text this makes it difficult to separate the contributions from the background and again it gives the paper a unstructured feeling. Minor remarks: Line 69: the authors say that the f_D(x) and y_D(x) are multivariate normal. Since they are uncountably infinite they are not proper multivariate normal distributions. They are Gaussian Processes instead. The same non-rigorous description of the posterior is at line 160. Line 160: "be be" Figure 6: It is my understanding that this batch parallelism is used to optimize the acquisition function. To properly evaluate the acquisition function parallelization wall-clock time the query of the GP is also to be factored in. This information is missing in the chart. Are you excluding the GP queries from the y axis? Why is the Hartmann6 function used for the feasibility constraints experiments and not some more common functions that come equipped with feasibility constriants? Why the comparison is only against random sampling? Line 338: "we give an specific example"

Correctness: The claim, methods and empirical methodology are correct.

Clarity: See above, I believe that the paper would benefit from some restructuring of the section following a more conventional NeurIPS writing style.

Relation to Prior Work: This is clear enough even if writing some of the sections in the background would make differentiating the contributions from the background easier to understand and clearer to varying level of readers.

Reproducibility: Yes

Additional Feedback:


Review 3

Summary and Contributions: This article presents result on the use of Sample Average Approximation for Bayesian optimization's acquisition functions in Monte Carlo form. In addition, a new way to compute the knowledge gradient acquisition function is proposed. Apart from these methodological and theoretical results, the paper details the capabilities of the BOTorch implementation.

Strengths: The paper provides an overview of available BO softwares, while mostly on BOTorch.

Weaknesses: Focusing a lot on the software part hides the methodological contributions (that are orthogonal), and is possibly uncommon. Assessments could be more balanced, and the evaluation of new results more exhaustive.

Correctness: The list of possible competitors is not complete (e.g., GPyFlowOpt is missing), it would be fairer to conduct an open competition. In fact there is a BO competition in NeurIPS 2020 Competition Track.

Clarity: The paper is easy to follow, but is perhaps too much focused on the software part.

Relation to Prior Work: The literature review is extensive.

Reproducibility: Yes

Additional Feedback: The benchmark results are relatively limited. More extensive tests could be performed e.g., with the COCO platform (Hansen, N., Auger, A., Mersmann, O., Tusar, T., & Brockhoff, D. (2016). COCO: A platform for comparing continuous optimizers in a black-box setting. arXiv preprint arXiv:1603.08785.) Another potential metric of interest is to show convergence based on wall time and not just the number of iterations. Efforts in installing GPflowOpt could be made, possibly by contacting the authors. Similarly many R packages provide BO capabilities. The drawbacks of the new KG formulation should be discussed. Overall a more balanced discussion on the methodology would be beneficial, to feel less promotional. *** Post-rebuttal comment *** I thank the authors for their response to my comments. As mentioned in the reviews, the mix of software, methodological and theoretical contributions in only 8 pages does not work very well. The additional 9th page may help a bit but not completely. For this reason, the paper is --in my opinion-- more suited for a software journal. The software capabilities are appealing, but the widespread use of this software will depend on many factors. I increased my score accordingly.


Review 4

Summary and Contributions: This work suggests the a sample average approach to the MC estimation of acquisition functions in Bayesian optimisation allowing the MC approximation of the acquisition function to be moved outside of the BO procedure, allowing for deterministic optimisers to be applied on the acquisition step. They provide theoretical convergence guarantees of this approach, and provide a complete software package to perform BO.

Strengths: The paper introduces a sample averaged approach to the MC approximation of intractable acquisition functions that allows for higher order optimisers to be used in BO instead of the first order stochastic gradient type approaches more traditionally used, and allows for the use of RQMC methods for variance reduction, these methods lead to faster convergence speeds and this is further supported by empirical results. The proposed SAA method is further supported by theoretical justification supporting the convergence of the approximated acquisition function to the true function, and the convergence of the maximiser to the true value along with an exponential convergence rate. The benefit of the SAA approach is well demonstrated in Section 6 where it is used to perform the evaluation of particularly challenging look-ahead acquisition functions such the KG acquisition functions. The method is accompanied by a well-designed modular software package that does have the potential to facilitate future research into Bayesian optimisation and acquisition functions which could prove useful to the BO community. Comparisons with serval existing software packages in this same area demonstrates that the introduced method provides performance gains over these implementations.

Weaknesses: This work provides two contributions, the advocation of the SAA of the acquisition function, and a software package to implement this approach, as well as other BO methods. While both aspects have their positives as discussed above it is not entirely clear that either aspect on its own is necessarily strong enough for acceptance. The authors mention many existing software packages that can implement the same models, and while BoTorch seems to be well written software with an appealing modular design and extensibility, it is not clear that this necessarily offers something fundamentally new to the ML community, even if it may prove useful. While the authors do argue that existing implementations lack their modularity or hardware acceleration, the relatively simple structure of the BO algorithm does mean that it can be implemented with relative ease using a functional approach inside of a hardware accelerated framework. Similar remarks hold when attempting to independently asses the contribution of the SAA approach, while there is some evidence of the benefits of this approach in Fig. 3, this figure is barely discussed in the main body of the paper. Again I would stress my generally positive opinions of the paper, but the presentation and frequent references to the appendix make distinguishing the standout contributions hard.

Correctness: The theoretical and methodological claims seem correct

Clarity: While in general sections of the paper are well written the overall paper does not seem to have been well condensed to the page limit as far too frequently references are made to material in the appendices. This significantly hurts the readability and flow of the paper, and importantly makes it harder to identify the standout contributions of the paper that were felt most important of inclusion in the main body. Additional minor comment, it is perhaps confusing to have $\alpha$ be both the general notation used for the acquisition function, and the exponential rate constant in Theorem 1.

Relation to Prior Work: Yes -- while there are numerous pre-existing software packages for BO the work in this paper has discussed these in good detail highlighting the main differences.

Reproducibility: Yes

Additional Feedback:

[Author Response · NeurIPS 2020]

We thank the reviewers for their insightful comments. As pointed out by R1, R2, and R4, our submission presents a distinctive framework for BO, with algorithms designed for modern hardware. Our work makes MC acquisition functions practical and easy-to-use by providing a modular design that carefully abstracts away a highly efficient computational system. Optimization within this framework is accomplished through a novel SAA approach backed by theoretical convergence results. We demonstrate the ease of use of our software—and the generality of our theory—with our parsimonious one-shot KG algorithm that achieves SoTA results. Our contributions have already enabled other BO researchers' novel methodological work in areas such as min-max optimization, optimization of risk measures, high-dimensional BO, transfer learning, and cost-efficient search. BoTorch has been applied in papers on material design, ML model reduction, drug discovery, trajectory optimization, and RL, to name a few. Our work is therefore a significant and timely contribution that expands the scope of problems that can be readily addressed through BO. We will use the extra page to address the review feedback and improve clarity, including adding details from the appendix.

We emphasize that the software and theoretical components in this case are highly *complementary*: to enable the widespread use of MC acquisition functions, BoTorch incorporates approaches such as fast predictive variance computations, parallelized posterior sampling, batch evaluation, and auto-differentiation of a large number of MC samples. Abstractions such as `fantasize` allow auto-differentiating through complex operations (in this case, batched low-rank updates of posterior variance caches and subsequent posterior predictions of conditioned GPs) in a transparent fashion. All of this functionality is specifically designed to exploit modern deep learning (DL) paradigms and hardware acceleration. Conversely, simply using a DL framework for conventional BO relying on Cholesky-based inference and analytic acquisition functions cannot effectively utilize the batch processing that DL frameworks are built around, and so will provide little benefit (this is a core aspect that distinguishes BoTorch from GPFlowOpt). Thus, our main contribution is the *joint* development of the MC+SAA methods together with a framework that renders their use practical, neither of which would, by itself, provide nearly as much value without the other.

Although submissions with different types of contributions can indeed be difficult to evaluate, they can also provide significant value that complements more traditional papers. There is a precedent at NeurIPS for domain-specific frameworks that leverage DL frameworks to simplify implementations and scale out computation (Gardner et al., 2018; Shazeer et al., 2018; Tran et al., 2018; Tran et al., 2019). For example, Tran et al. [2018] describe embedding probabilistic programming into DL frameworks, and Tran et al. [2019] discuss a collection of "layers" that intertwines DL components with probabilistic elements. Like our work, these papers provide a framework for research that abstracts away computational aspects to simplify and scale computation. We additionally provide theoretical results that form the basis of new approaches to BO.

**R1: (1)** Our motivation was to clearly lay out the abstractions and how they fit with the MC framework before illustrating their use. Thanks for your suggestions to improve the clarity of these sections, which we will incorporate. **(2)** The pending points strategy and the decorator are described on pg 5, in the paragraph directly before Sec 5.1. **(3)** We will be more specific in our discussion about GPFlowOpt, and hope the main text above clarifies this.

**R2: (1)** To provide better background, we will introduce aspects of the work such as fantasies and KG earlier on. **(2)** $f_D(\mathbf{x})$ and $y_D(\mathbf{x})$ are the posterior at the set of points $\mathbf{x}$, so under a GP prior these are indeed MVNs. The abstract posterior objects $f_D$ and $y_D$ are of course GPs. We will make this distinction more precise. **(3)** Wall times in Fig. 6 are end-to-end, including querying the GP. **(4)** The constrained experiments compare against random since (to our knowledge) other maintained packages do not support constraints. We used H6 to contrast with unconstrained results.

**R3: (1)** The goal of our work is to present a practical and efficient way of doing BO with MC acquisition functions. To this end, we demonstrate our methods compare favorably across a variety of well-established packages—to the satisfaction of other reviewers, and in line with the BO literature published at NeurIPS and ICML. **(2)** We do reference GPyFlowOpt in Sec 2. Given that GPFlowOpt (i) has never been rigorously evaluated (including within the GPFlowOpt tech report itself), (ii) has an algorithmic foundation that is not amenable to MC acquisitions, (iii) has not been maintained for 2 years, and (iv) won't build; we hope the reviewers will agree that comparison is not critical. **(4)** In Fig. 12, we included wall time comparison between KG implementations. Additionally, here are average wall time (sec) per iteration for CPUs across the synthetic problems (Figs 4, 10, 11, 13): MOE EI, 0.1; BoTorch EI, 1.1; BoTorch NEI 9.0; BoTorch OKG, 102.3; Dragonfly, 141.4; MOE KG, 160.6. We did not record per-iteration runtimes of GPyOpt (due to its full-loop implementation), but can include results in the camera-ready version. Note that MOE EI performed poorly on all benchmarks. **(5)** We will clarify the drawbacks of OKG. The primary drawback is that the dimensionality of the optimization problem grows with the number of fantasies $N_f$. In addition, memory complexity, due to batched GPs used for fantasization, also grows with $N_f$. In practical settings, only a moderate $N_f$ can be used (we use 128).

**R4: (1)** While some BO algorithms do have simple structure, this is not universally true (NEI, KG). Further, the fact that BO can be implemented in a functional way using hardware acceleration does not mean that it will be particularly fast or easy to work with (see above). **(2)** We will use the additional page in the final version to move relevant parts of the appendix into the main paper, including the SAA results.

[Meta-Review · NeurIPS 2020]

This paper presents BOTorch, an efficient Bayesian optimization library that enjoys several advantages over existing ones, such as a novel approach to optimize MC acquisition functions using fixed sample averages, a faster and easier implementation of second-order methods and look-ahead BO methods (e.g., Knowledge gradient), and general convergence results for sample-average approximation to acquisition functions via randomized quasi-MC. Overall, a good paper with strong results.